# Occurrence, Diversity and Anti-Fungal Resistance of Fungi in Sand of an Urban Beach in Slovenia—Environmental Monitoring with Possible Health Risk Implications

**DOI:** 10.3390/jof8080860

**Published:** 2022-08-16

**Authors:** Monika Novak Babič, Nina Gunde-Cimerman, Martin Breskvar, Sašo Džeroski, João Brandão

**Affiliations:** 1Department of Biology, Biotechnical Faculty, University of Ljubljana, Jamnikarjeva 101, 1000 Ljubljana, Slovenia; 2Department of Knowledge Technologies, Jožef Stefan Institute, Jamova Cesta 39, 1000 Ljubljana, Slovenia; 3Jožef Stefan International Postgraduate School, Jamova Cesta 39, 1000 Ljubljana, Slovenia; 4Department of Environmental Health, National Institute of Health Dr. Ricardo Jorge, Av. Padre Cruz, 1600-609 Lisbon, Portugal; 5Centre for Environmental and Marine Studies (CESAM)—Department of Animal Biology, University of Lisbon, Campo Grande, 1749-016 Lisbon, Portugal

**Keywords:** beach sand, fungal diversity, health, influence of environmental factors, leisure activities, monitoring of sand, resistance to antimycotics, urban beach

## Abstract

Beach safety regulation is based on faecal indicators in water, leaving out sand and fungi, whose presence in both matrices has often been reported. To study the abundance, diversity and possible fluctuations of mycobiota, fungi from sand and seawater were isolated from the Portorož beach (Slovenia) during a 1-year period. Sand analyses yielded 64 species of 43 genera, whereas seawater samples yielded 29 species of 18 genera. Environmental and taxonomical data of fungal communities were analysed using machine learning approaches. Changes in the air and water temperature, sunshine hours, humidity and precipitation, air pressure and wind speed appeared to affect mycobiota. The core genera *Aphanoascus*, *Aspergillus*, *Fusarium*, *Bisifusarium*, *Penicillium*, *Talaromyces*, and *Rhizopus* were found to compose a stable community within sand, although their presence and abundance fluctuated along with weather changes. *Aspergillus* spp. were the most abundant and thus tested against nine antimycotics using Sensititre Yeast One kit. *Aspergillus niger* and *A. welwitschiae* isolates were found to be resistant to amphotericin B. Additionally, four possible human pollution indicators were isolated during the bathing season, including *Meyerozyma*, which can be used in beach microbial regulation. Our findings provide the foundations for additional research on sand and seawater mycobiota and show the potential effect of global warming and extreme weather events on fungi in sand and sea.

## 1. Introduction

The fungal kingdom includes highly diverse group of ubiquitous organisms with heterotrophic metabolisms [1], obtaining energy from simple sugars to long-chained and cross-linked compounds, such as cellulose, lignin, polycyclic hydrocarbons, and even human-made plastic materials [2]. Fungi need oxygen to decompose organic matter; thus, they are usually present in water, and aerated layers of soil in all geographical regions [1,2]. On the other hand, some genera and species, described as extremophiles, have been isolated also from samples of rocky and sandy deserts [3,4], salterns [5,6], and brine [7]. Extremophiles do not only survive but are also able to propagate in habitats with combined harsh living conditions, such as high salinity, low water activity, prolonged UV-radiation, drought, and high or low temperatures [8,9]. Living conditions in many niches in nature often alternate, leading to selection of extremotolerant organisms [10]. One such habitat is beach sand. Like other soils, beach sand contains organic matter, originating from microorganisms, plants, animals, human activities, and the sea [11]. Water activity in the beach sand can be frequently low, sand can be exposed to prolonged UV-radiation and presence of salts [12,13]. The combinations of these factors contribute to the overall microbial load in beach sand. In the past, attention was mainly specific in the contest of health-related bacteria and viruses, particularly those causing gastrointestinal illnesses [14,15]. Recently, fungi in beach sand became a frequent subject of interest, especially as some detected taxa cause dermatomycoses, allergic reactions, otitis, and deep mycetoma [16]. Opportunistic environmental fungi came under scrutiny also due to the evolution of their genes encoding resistance to antimycotics, as recognised for the ubiquitous genera *Aspergillus*, *Candida*, and *Fusarium*, also regularly isolated from sand [17]. Thus, beach sand may pose a certain health risk for people spending a lot of time in close contact with it [13]. So far it has been recognised that beach sand harbours a great diversity of fungi including dermatophytes, dematiaceous fungi, plant- and insect-related fungi, halotolerant and xerotolerant fungi, and fungi related to anthropogenic pollution [13]. The filamentous genera *Aspergillus*, *Cladosporium*, *Chrysosporium*, *Curvularia*, *Fusarium*, *Mucor*, *Penicillium*, *Rhizopus*, *Scedosporium*, *Scopulariopsis*, *Scytalidium*, *Stachybotrys*, and *Trichophyton* usually prevailed [13,18,19,20] over the yeasts and yeast-like fungi *Aureobasidium*, *Candida*, *Geotrichum*, *Exophiala*, *Metschnikowia*, *Rhodotorula*, and *Yarrowia* [12,21]. Nevertheless, despite the diverse scientific approach and identification methods used, still little is known about the general structure of fungal communities in beach sand, in particular the identification of main core-genera and factors influencing their fluctuations, as well as the presence of possible indicator species, promoted after certain events. In addition, little is also known about the presence of fungicide-resistant strains colonising beach sand. 

As a follow-up to the white paper “Beach sand and the potential for infectious disease transmission: observations and recommendations” [12], 13 countries initiated the European “Mycosands” project, with the general aim to shed some light onto fungal diversity in beach sands, as well as to understand the possible health risk they could pose to humans [13]. The initiative was limited to culture-specific approaches for the isolation and characterisation of fungi. The present study in Slovenia was conducted as a part of that initiative, during which we sampled the beach sand on an artificial beach that has no direct contact with seawater. 

Samples were taken at the Central Portorož beach, located in the North Adriatic Sea, in the Gulf of Trieste, and subjected to the weather conditions of the Mediterranean Sea. This sandy beach, which was created artificially, was chosen for the study because it is one of the most popular beaches in Slovenia: This is due to its location in the center of an urban environment, surrounded by buildings, a main road and vegetation. Sand on the surface is regularly cleaned and mechanically aerated, starting a month before and lasting during the complete official bathing season (between June and September). The beach regularly receives the Blue Flag Award, a quality certification for being clean, safe and user-friendly.

Sampling was performed monthly during a 1-year period. For the monitoring, we focused particularly on environmental factors possibly affecting the presence of fungal taxa. Additionally, our aim was to determine whether sand harbours culturable fungi that could be associated with the presence of humans or certain environmental events (indicator fungi). Finally, as the most abundant fungi, the *Aspergillus* strains obtained during the official bathing season were tested against nine antifungals to determine possible resistance to antimycotics. 

## 2. Materials and Methods

### 2.1. Sampling of Sand and Seawater

Sampling was conducted at the Central beach in Portorož, Slovenia, as a part of the wider European initiative “Mycosands” [13]. Sampling of the quartz sand and seawater was performed monthly between October 2018 and September 2019. Supratidal sand was sampled between 9 am and 10 am at a depth of 5–10 cm as pre-arranged in the Standard Operational Procedure (SOP). The SOP was adopted from Sabino et al. [19] and agreed among the leaders of the Mycosands project [13]. The sand was aseptically collected into sterile 50 mL Falcon flasks, and later homogenised in sterile plastic bags. The total weight of the final samples was ~200 g. Seawater samples were collected aseptically into 2 sterile 500 mL vessels (Golias, Ljubljana, Slovenia) at a depth of 20 cm in a 1 m-deep water column. All samples were labelled and transported to the laboratory in a cooler within 2 h of sampling, as described by Sabino et al. [19].

### 2.2. Cultivation of Fungi and Storage of Viable Strains

Forty grams of a crude sand sample were placed into sterile Erlenmeier flasks. Fungi were extracted from the sand with the addition of 40 mL of sterile distilled water and shaking for 30 min at 100 rpm. A total of 200 µL of the obtained sand extract and 200 µL of the collected seawater were plated in triplicate onto Sabouraud’s Dextrose agar (SDA) (Biolife, Milano, Italy) and Mycosel agar (Becton Dickinson, Heidelberg, Germany) with the addition of Cycloheximide and Chloramphenicol in order to prevent the overgrowth of bacteria. SDA plates were incubated for 5 to 7 days, and Mycosel plates were incubated for up to 21 days to also obtain slow-growing dermatophytes. Both media were incubated at 25, 30 and 37 °C. Additionally, 5 mL of sand extract and 20 mL of seawater samples were filtered using Millipore (Merck, Darmstadt, Germany) filters with a pore diameter of 0.45 µm. Filters were aseptically placed onto Malt Extract Agar (Biolife, Milano, Italy) supplemented with 10% (*v/w*) of NaCl and Chloramphenicol (0.05 g per liter). Plates were incubated for 5 to 7 days at 25, 30 and 37 °C for cultivation of halotolerant species. 

After the incubation, morphotypical colonies were counted, and colony-forming units (CFU) per gram of crude sand and per liter of seawater were counted and finally reported as the average number of triplicates. Each of the morphotypes were then transferred to fresh SDA, Mycosel, or MEA + 10% NaCl and incubated for 5–20 days until visible growth.

All strains obtained in the study were permanently stored in the Ex-Culture Collection of the Infrastructural Centre Mycosmo, MRIC UL, Slovenia (http://www.ex-genebank.com/ (accessed on 16 August 2021)), at the Department of Biology, Biotechnical Faculty, University of Ljubljana.

### 2.3. DNA Extraction and Molecular Identification

Cultures were grown either on MEA or on Mycosel plates, until visible growth of mycelium (usually 5–7 days). DNA of filamentous fungi was extracted according to the protocol of Van den Ende and de Hoog [22] using mechanical lysis and chloroform extraction. Yeasts were grown on MEA for 3–5 days, before DNA was extracted with PrepMan Ultra reagent (Applied Biosystems, Foster City, California, USA) following the manufacturer’s instructions. The DNA samples obtained were stored at −20 °C prior their use in the identification process.

Fungi were identified by observing their macro- and micromorphological features. According to these, the final identification of filamentous fungi based on the rDNA nucleotide sequences of either the partial actin gene (*act*), the partial beta-tubulin gene exons and introns (*benA*), or the partial translation elongation factor 1-alpha (*tef 1-α*) and the whole internal transcribed spacer region (ITS = ITS1, 5.8S rDNA, ITS2). Identification of the yeasts was conducted based on the sequences of the large subunit ribosomal DNA (LSU = partial 28S rDNA, D1/D2 domains). The primers used for the amplification and sequencing were ACT-512F and ACT-783R (*act*) [23], Bt2a and Bt2b (*benA*) [24], EF1 and EF2 (*tef 1-α*) [25], ITS5 and ITS4 (ITS) [26], and NL1 and NL4 (LSU) [27]. Sequencing was performed at Microsynth AG, Austria. Sequences were obtained at and assembled by FinchTV 1.4 (Geospiza, PerkinElmer Inc., Seattle, Washington DC, USA) and adjusted with the Molecular Evolutionary Genetics Analysis (MEGA) software version 7.0 [28]. Identification of sequences was carried out using the BLAST algorithm on the NCBI web page [29] and finally compared with type strains from taxonomical database on Westerdijk Fungal Biodiversity Institute (Utrecht, The Netherlands). Fungal names were checked in Index Fungorum (www.indexfungorum.org (accessed on 16 August 2021)), and the sequences of representative strains obtained in the study were deposited in the GenBank database (NCBI).

### 2.4. Antifungal Susceptibility Testing of Selected Aspergillus Strains

Antifungal susceptibility testing of 10 *Aspergillus* strains was performed with the YeastOne YO10 Kit (ThermoFisher Scientific, Waltham, Massachusetts, USA) containing 9 antifungal agents: Amphotericin B (AB, dilution range 0.12–8 µg/mL); Caspofungin (CAS, dilution range 0.008–8 µg/mL); Fluconazole (FZ, dilution range 0.12–256 µg/mL); Itraconazole (IZ, dilution range 0.015–16 µg/mL); 5-Flucytosine (FC, dilution range 0.06–64 µg/mL); Voriconazole (VOR, dilution range 0.008–8 µg/mL); Anidulafungin (AND, dilution range 0.015–8 µg/mL); Micafungin (MF, dilution range 0.008–8 µg/mL); and Posaconazole (PZ, dilution range 0.008–8 µg/mL). The suspensions of conidia were diluted with YeastOne broth according to the manufacturer’s instructions to reach final inoculum between 0.5–5 × 10^4^ CFU/mL. Each well of the dried YeastOne plates was rehydrated with 100 μL of the fungal suspension. The inoculated panels were incubated at 35 °C, and read after 24 and 48 h. Growth in the wells was indicated by color change from blue (negative) to pink (positive), and the minimal inhibitory concentration (MIC) was determined as the first concentration at which the growth indicator remained negative.

### 2.5. Using Machine Learning to Relate Environmental Conditions and Fungal Profiles

The data collected during 12 months of sampling were split into two distinct kinds. Variables of the first kind describe the environment and variables of the second kind describe the presence of fungi, i.e., the sampled fungal profiles. The variables of the first kind are the following: season, month, average air temperature (°C), maximal air temperature during the past 7 days before sampling (°C), average soil temperature at the depth of 5 cm (°C), average air humidity (%), air humidity in the sampling day (%), monthly rainfall rate (mm^3^), monthly sunshine hours (h), average daily sunshine hours (hh:mm), average air pressure (hPa), average wind speed (m/s), sea water temperature (°C) (data was obtained from ARSO [30] during 2018 and 2019), weather (sunny, dry, hot, cloudy, humid, rainy, foggy, cold, windy), unexpected events (flood, strong wind, heatwave, open celebration, construction work), bathing season, beach cleaning (which includes picking up waste and mechanical aeration), characteristics of beach sand (wet—coastal bottom sediment, humid—vadose zone sand, dry—supratidal sand), and pH (seawater, sand extract). The variables of the second kind record the presence and abundance of fungal species and taxonomical groups (genera, phylum, etc.), as well as functional/non-taxonomical fungal groups (human colonisers, dermatophytes, etc.). Detailed descriptive and numerical data of all variables are included in the Appendix A.

### 2.6. Predictive Modelling

Predictive Clustering Trees (PCTs) [31] are a generalised version of standard decision trees belonging to the class of predictive models constructed by machine learning (ML). A decision tree of this kind can be viewed as a hierarchy of non-overlapping clusters. The PCT induction algorithm recursively splits the initial data (a large non-homogeneous cluster) into smaller, purer (more homogeneous) clusters. The splits (testing conditions) within the decision tree are selected by calculating the values of a heuristic function (the intra-cluster variance of the target variables) and maximising the reduction of its value: this reduction of variance can be also used to estimate variable importance in PCT ensemble approaches [32,33]. Many data splits are considered and those with the best heuristic values are the ones that are chosen to appear in the decision tree. Such data partitioning continues until a stopping criterion is met. Nodes at the bottom of the decision tree (technically, very small clusters) are called leaf nodes and provide predictions for the values of the target variables.

PCTs can be parametrised to address several ML tasks. Two of those tasks were addressed in our analysis, namely multi-target regression (MTR) and multi-label classification (MLC). We first declared specific fungi (and higher taxa from the corresponding taxonomy) and functional groups as labels (target variables with binary values) and trained MLC models to predict their presence/absence in the sand and water samples. We then consider the abundances (CFU/g) of the functional groups as real-valued targets and address the corresponding MTR task. Each of the three figures depicts one predictive model (a PCT) with leaf nodes giving predictions for multiple targets of interest, i.e., presence and abundance of different groups of fungi.

PCTs were trained using all available input features and were allowed to grow until all leaf nodes contained 2 and 4 examples (pre-pruning). The quality of the trained predictive models was evaluated by calculating appropriate metrics of predictive performance. Specifically, for the MTR setting, we calculated the relative root mean squared error (RRMSE) and for MLC setting, the weighted area under the precision recall curve (AUPRC). Since we do not have a separate test set, the predictive performance was estimated with a leave-one-out cross validation procedure.

## 3. Results

### 3.1. Artificially Designed Sandy Beach Harbours a Great Diversity of Fungi

A 1-year-long survey with monthly sampling of beach sand provided insights into the diversity of culturable fungal species constituting the fungal community. Altogether, fungi isolated from the beach sand were classified into 43 genera and 64 species. The number of different genera isolated from sand in the spring varied between 10 and 13 (average = 11), in the summer 11–13 (average = 12), in the autumn 6–13 (average = 10), and in the winter 4–13 (average = 8). Two months stood out: in November, the number of genera dropped to 6 in comparison to September and October, and in January it rose in comparison to December and February to 13. The distribution and diversity of all identified genera over the period of 12 months is shown in Figure 1. Among them, 10 species, namely *Actinomucor elegans*, *Aspergillus flavus*, *A. lentulus*, *A. terreus*, *Candida tropicalis*, *Fusarium solani*, *Microascus brevicaulis*, *Phialophora americana*, *Sarocladium kiliense*, and *Trichosporon asahii* belonged to the microorganisms of Biosafety Level 2 (BSL-2) (Table 1).

### 3.2. Differences in Diversity of Culturable Fungi in Seawater and Beach Sand

We detected 18 genera and 29 different species of fungi in seawater samples. Amongst them, only *Aphanoascus fulvescens* and *Wickerhamomyces anomalus* were recognised as BSL-2 fungi. Interestingly, these neighboring environments had only three species in common (e.g., *Aspergillus tubingensis*, *Cladosporium halotolerans*, and *Penicillium crustosum*). The greatest diversity in the fungal community of seawater samples was observed in February, June, and October, with 5, 5, and 8 isolated genera respectively. The highest diversity was observed in October, with elevated numbers of the genus *Cladosporium*. During the same period, most *Aspergillus* species were isolated, while genera *Aureobasidium*, *Paraconiothyrium*, and *Penicillium* occurred during the summer and autumn. In addition, black yeasts (e.g., *Exophiala* and *Hortaea*) and human-related yeasts *Meyerozyma* and *Wickerhamomyces* were also isolated during the late summer and autumn months. However, their presence was only sporadic, which possibly indicates transient human pollution episodes. 

### 3.3. Machine Learning Models Reveal Connections between Environmental Changes and Fungal Presence

Each fungal species was assigned into at least one group namely dermatophytes, dematiaceous fungi, rock-inhabiting fungi, halotolerant and halophilic fungi, psychrotolerant fungi, fungi related to plants, soil, and freshwater, fungi related to insects, BTEX (as benzene, toluene, ethylbenzene, and xylene) degraders, possible indicators for human presence, and the “core-fungi” of the sand (Table 1). 

Figure 2 depicts a Predictive Clustering Tree (PCT) model that highlights the habitat, e.g., sand or seawater, as the first (most important) decisive factor that discriminates between both fungal communities. When looking into the fungal community of the sand (left section of the PCT depicted in Figure 2), the core genera, psychrotolerant fungi, dematiaceous fungi, and halotolerant/halophilic fungi are present in every leaf, regardless of the screened environmental factors. All non-taxonomical groups together are likely to be present in sand when the monthly rainfall rate is ≤28.5 mm^3^ (March, June, August, and January). Among them, rock-colonising genera appeared only under these conditions. Fungi related to the bathing season, human colonisers, dermatophytes, insects’ colonisers and rock colonisers are absent during the months when the monthly rainfall rate is >28.5 mm^3^ and the monthly sunshine hours are ≤163.4 h (May, November, and December). The last decision step following the monthly rainfall rate >28.5 mm^3^ and monthly sunshine hours ≤163.4 h is the season, which predicts the absence of BTEX-degrading fungi during the autumn and winter months, from September to February.

For fungal communities from seawater, pH is the first decision factor. The leaf with pH > 8.11 rules out BTEX degraders and dematiaceous fungi, while fungi related to the bathing season and rock-colonising fungi appear only during the autumn months when pH ≤ 8.11. Human-colonising fungi appeared in three leaves, independent of the pH of the water, yet their presence was influenced by the season, air humidity, and daily sunshine minutes. They can be present in seawater when pH > 8.11 (August, September), but can be isolated also when pH ≤8.11 during the autumn months (October, November), or when pH ≤8.11 in the other seasons, with average air humidity >66% and when average daily sunshine minutes exceeds 587 min (April, May). The group of halotolerant and halophilic fungi is present in every tree leaf, independent of all environmental factors included in the tree. On the other hand, the groups of fungi related to insects and dermatophytes did not appear in any of the tree leaves.

### 3.4. Factors Influencing the Stability of Fungal Community in Beach Sand

Fungal genera isolated from more than seven sand samples (>7/12) were recognised as the “core-genera”, representing the backbone of fungal community in the sand. They include the species of *Aphanoascus fulvescens*, *A. reticulisporus*, *A. terreus*, *Aspergillus calidoustus*, *A. destruens*, *A. flavus*, *A. floccosus*, *A. jensenii*, *A. lentulus*, *A. nidulans*, *A. niger*, *A. pseudoglaucus*, *A. terreus*, *A. tubingensis*, *A. welwitchiae*, *Fusarium brachygibbosum*, *F. ipomoeae*, *F. nirenbergiae*, *F. solani*, *Bisifusarium delphinoides*, *Penicillium aurantiogriseum*, *P. crustosum*, *P. glabrum*, *P. griseofulvum*, *P. oxalicum*, *P. sizovae*, *Penicillium* sp., *Talaromyces atroroseus*, *T. pinophilus*, *T. purpureogenus*, and *Rhizopus arrhizus* (Table 1, Figure 3). Although their numbers fluctuated over time, their presence was stable in comparison to the other genera in the community.

The machine learning-generated MTR tree describes the highest numbers of the genera *Aphanoascus* (19.2 CFU/g), *Fusarium & Bisifusarium* (510.0 CFU/g), and *Rhizopus* (201.7 CFU/g) in months when the seawater temperature was >9.9 °C, average air pressure >1012.0 mbar, average wind speed >2.7 m/s, and monthly rainfall rate ≤105.7 mm^3^ (August and October) (Figure 4). On the other hand, following the same environmental factors, the presence of *Aspergillus* and *Penicillium* was the lowest, with 252.8 CFU/g and 15.0 CFU/g respectively. The tree describes the highest *Penicillium* numbers (118.0 CFU/g) when seawater temperature was >9.9 °C and average air pressure ≤1012.0 mbar (May, July). The highest number of *Aspergillus* in beach sand is expected when seawater temperature is ≤9.9 °C (January–March) (Figure 4).

### 3.5. Potential Human Pollution Indicators for the Bathing Season

During the yearly monitoring, we observed a short-term elevated presence of four genera, including five species: *Actinomucor elegans*, *Condenascus tortuosus*, *Meyerozyma caribbica*, *M. guilliermondii*, and *Stachybotrys* sp. Their numbers increased mainly from May to October and were particularly elevated during the official bathing season from June to September. As their occurrence and increased number coincided with the official time of the bathing season, they could be used as potential human pollution indicators (Table 1, Figure 5).

Figure 6 depicts a PCT model that predicts the abundance of the four genera based on environmental factors. Maximal air temperature during the past 7 days before sampling was identified as the most discriminating environmental factor. When its values are higher than 28 °C, the presence of all four indicator genera in the beach sand is predicted. This value corresponds with the summer season and the official bathing season. However, these indicator genera may be present in sand in lower numbers also outside the bathing season. The genera *Condenascus* and *Stachybotrys* were linked to the air pressure ≤1012.0 mbar, while *Meyerozyma* appears in autumn when the air pressure is >1012.0 mbar. *Actinomucor* is most likely isolated from sand when air pressure is >1012.0 mbar, during the whole year, except for the autumn months (Figure 6).

### 3.6. Aspergillus Niger Strains Isolated from Beach Sand Are Resistant to Amphotericin B

Fungi of the genus *Aspergillus* were with the monthly average number of 376 CFU/g also the most represented fungi in the beach sand samples. In total, 10 strains of *Aspergillus* spp, isolated during bathing season, were subjected to the antifungal susceptibility testing in order to evaluate possible risk for humans during their leisure activities on the beach. The detailed results carried out with Sensititre Yeas One Kit are summarised in Table 2.

The minimum inhibitory concentration (MIC) for echinocandines, after 24 h was as follows: anidulafungin = 0.03 µg/mL, except for *A. welwitschiae* (0.015 µg/mL), micafungin = 0.06 µg/mL for *A. flavus* and *A. welwitschiae*, and 0.12 µg/mL for *A. nidulans* and *A. niger*. *Aspergillus nidulans* was the most susceptible to caspofungin (MIC = 0.25 µg/mL). After 48 h, MIC of echinocandines for all *Aspergillus* species was >8 µg/mL as observed by Siopi et al [36]. 

MIC for amphotericin B and 5-flucytosine for *A. flavus* and *A. nidulans* were 2–4 µg/mL and >64 µg/mL, respectively. These species are known to be intrinsically resistant to amphotericin B as reported by European Committee on Antimicrobial Susceptibility Testing (ECOFF = 4 µg/mL). Among the tested strains, only *Aspergillus niger* was resistant to amphotericin B (MIC = 2 µg/mL vs. ECOFF = 1 µg/mL). Additionally, its taxonomically close relative *A. welwitschiae* (section Nigri) yielded the same result (MIC = 2 µg/mL), but there is so far no known ECOFF data for this species. 

Regarding tested triazoles, all *Aspergillus* species were the most susceptible to posaconazole and itraconazole, while all were resistant to fluconazole with MIC ≥ 256 µg/mL.

## 4. Discussion

Fungi as heterotrophs have crucial role in the circulation of organic matter in the environment. Many species are known to grow under elevated UV-radiation, desiccation and salinity, and break down long-chained hydrocarbons such as chitin, cellulose, lignin and crude oil, as well as on human-made materials, such as rubbers, silicone and different types of plastic [9,10,38,39]. Human-made materials are increasingly represented in the environment with a growing population, and frequently end up in the oceans [40,41]. Beach sand and sea bordering urban places usually meet many of the above-described conditions as explained in detail in the white paper “Beach sand and the potential for infectious disease transmission: observations and recommendations” [12]. During the last few decades, fungi have become frequently connected also to a variety of human diseases [34,42]. Contrary to viral or bacterial infections, the symptoms of fungal infections usually appear later and are therefore more difficult to relate to the primary environmental source [34,43]. Studies on fungal taxonomy and diversity in different habitats are thus important in order to fill the environmental gap in medical data. One of such studies is the European initiative “Mycosands”, investigating fungal diversity and abundance in beach sand and seawater, under different environmental conditions in order to assess the possible health risk fungi could pose to humans [13]. 

Compared to the other participating countries, Slovenia’s location possesses some geographical and natural specifics, for instance higher salinity and elevated concentrations of mercury [44]. Due to the location, sea flows, and the close proximity of major ports, pollutants such as petroleum may have been present [45]. The sampling site, the Central Beach Portorož, is a man-made beach with fine quartz sand, located in the northern Adriatic Sea (Gulf of Trieste) as a part of a touristic center, surrounded by buildings and a main traffic road with vegetation along the road; thus, we classified it as an “urban beach” [13]. Another characteristic is the concrete footpath, separating the sand from the sea, preventing the tidal wash-off of the sand. A month before and during the official bathing season (May–September), employees regularly pick up waste and mechanically aerate the sand. In addition, the beach regularly receives the Blue Flag Award as a synonym for its quality as a clean, safe, and user-friendly beach [46]. However, microbiological monitoring of the beach sand is not a part of the Bathing Water Directive [47], though the most recent World Health Organization guidelines for recreational water quality mention fungi and the risk assessment of non-generic parameters, when relevant [48]. To the best knowledge of the authors, this is the first report of fungal presence and diversity in beach sand and related seawater in Slovenia. During the monitoring, we identified 64 species from the sand, and 29 species from seawater samples. The results are comparable with previous global studies on sand, reporting a high diversity of fungi, even when sand is dry, exposed to the sun, and well-aerated [49,50,51]. 

### 4.1. Fungi Isolated from Seawater Differ from Sand Mycobiota

Interestingly, the results showed significant difference between the mycobiota of seawater and beach sand. Both habitats had only three species in common (i.e., *Aspergillus tubingensis*, *Cladosporium halotolerans*, and *Penicillium crustosum*). This difference was identified also with machine learning analysis, where the habitat appeared as the first, most decisive, factor (Figure 2). Besides salinity, UV radiation, temperature fluctuations and water flow, the concrete footpath between both environments could also affect the fungal biota, preventing the washing off of the sand and its possible mixing with seawater. Seawater samples expectedly yielded fewer fungal taxa in comparison to the sand [13], with only two species classified as BSL-2. The elevated number of taxa in seawater corresponded to high rainfall rated in February and June and flooding in October. This is particularly visible for the genera *Aspergillus* and *Cladosporium* isolated during these months, likely due to the washout from land [52]. In late summer and autumn, we isolated the melanised genera *Aureobasidium*, *Exophiala*, and *Hortaea* known to be well adapted to UV-irradiation and salinity, both of which are elevated during the late summer months [44,53]. Human-related yeasts *Meyerozyma* and *Wickerhamomyces* were expectedly isolated during this period, due to the official bathing season and presence of people [54], although their number in seawater was low. 

According to the machine learning analysis, the pH is the first, most decisive, factor affecting the fungal community in seawater. The Northern Adriatic Sea has the lowest pH during the winter months and the highest during late spring and summer [55]. Our local measurements showed the peak of pH during the summer and early autumn, and its lowest values from October until June. The months with the highest pH (>8.11) yielded less fungi and particularly lacked dematiaceous, melanised fungi and BTEX degraders. The reason for this could be the negative effect of alkaline pH on pigment formation in certain genera [56]. More fungi were isolated from seawater with pH ≤ 8.11 during the autumn months, in particular fungi associated with the bathing season and rock-colonising fungi. In comparison to the sand samples, these groups appear in the seawater a month or two after their presence is noted in sand. 

However, the pH of seawater does not affect the presence of human-colonising fungi, which is most related to the season, air humidity, and daily exposure to sunshine. They appeared in the seawater sporadically from August to November, but were present also in moist and sunny months in spring. This observation is in accordance with Kaewkrajay et al. [57], reporting the presence of the genera *Candida*, *Meyerozyma*, *Rhodotorula* and *Wickerhamomyces* in seawater [57]. Again, the time lapse of a month or two is visible when compared to the sand samples where these groups appear from June to October. The reason for the time lapse could be the precipitation and washout from the sand [58]. 

It is noteworthy that the group of halotolerant and halophilic fungi was present during the whole year, independently of environmental factors included in the study. These results were expected, since the seawater contains a stable community of halotolerant genera [53]. In addition, the models predict that the insect-related fungi and dermatophytes are not present (Figure 2), confirming their close connection to the terrestrial hosts. 

### 4.2. Fungal Abundance in Beach Sand Varies According to Seasonal Weather Conditions

Monthly monitoring of the beach sand revealed the influence of seasonal weather on the fungal community. The highest fungal diversity in beach sand was described when the monthly rainfall rate was ≤28.5 mm^3^. Since the rock-colonising genera were also present, this suggests a possible washout of fungi during the rainy periods and a reestablishment of the community during the dry periods [58,59,60]. In months when the monthly rainfall rate was higher than 28.5 mm^3^ and consequently fewer sunshine hours were counted (≤163.4 h), fungi found during the bathing season, such as human colonisers, dermatophytes, insect-related fungi, and rock colonisers were absent in the beach sand. This may suggest both the washout of fungi and also the absence of hosts, e.g., humans and insects [13,61,62]. In addition, fungi reportedly connected to degradation of BTEX compounds are absent in the months from September to February, when the monthly rainfall rate is higher than 28.5 mm^3^, and the sunshine hours are lower than 163.4 h. This could be attributed to the winter season: lower temperatures, reduced beach activity, and less traffic on the roads and the sea nearby, with consequently less oil and gas pollution [63]. 

While there were differences observed for the above-mentioned fungal groups, the machine-learning model (PCT depicted in Figure 2) pointed out the presence of the core genera, psychrotolerant, dematiaceous, and halotolerant fungi in the beach sand regardless of the screened environmental factors. The results are expected, since sand is a known reservoir of *Aspergillus*, *Penicillium*, *Cladosporium*, *Fusarium*, *Rhizopus*, and dermatophytes [13,18]. The results relate fungal growth in the environment to longer sun exposure and moderate temperatures [64]. However, changes were noted in November, when the number of isolated genera dropped significantly and in January, when it elevated. The likely reason for lower diversity in November is the drop of monthly sunshine hours. This factor was previously reported as one of the most influencing for microbial communities in sand due to its double effect on daily temperatures and UV-irradiation [12,13]. 

On the other hand, the reason for higher fungal diversity in January sampling was a one-time event, the New Year celebration on the beach, which resulted also in the detection of human-related genera *Candida* and *Rhodotorula* [13,51,65]. In addition, 5 out of 10 species (*Aspergillus flavus*, *A. terreus*, *Candida tropicalis*, *Fusarium solani*, and *Microascus brevicaulis*) recognised as human opportunistic pathogens (BSL-2) [34] were isolated in January. This leads to the conclusion that one-time events could also significantly affect fungal abundance and diversity in beach sand. 

### 4.3. Core Fungal Genera in the Sand Are Affected by the Seawater Temperature

During the study, we particularly focused on the constantly present core-genera and possible indicator fungi in sand samples, appearing mainly during the bathing season. The core-genera include the dermatophyte *Aphanoascus*, *Aspergillus*, *Fusarium*, *Bisifusarium*, *Penicillium*, *Talaromyces*, and *Rhizopus*. They usually colonise plants and plant debris, as well as other organic materials, and are often isolated from soil, dry beach sand, deserts, and dust [13,66]. Particular attention should be paid to the elevated numbers of thermotolerant species such as *Aspergillus fumigatus*, *A. terreus*, *A. glaucus*, *A. nidulans*, *A. niger*, *A. sydowii*, *Fusarium solani*, *Rhizopus arrhizus,* and dermatophyte *Aphanoascus fulvescens*, since these are frequently involved in a variety of fungal infections and can represent health hazards for beach-goers [18,34]. Despite forming stable fungal communities in the sand, the numbers of these genera fluctuated over time. The machine learning method found seawater temperature to be the first and most decisive factor in the changes of the core mycobiota (Figure 4). The changes in seawater temperature are slower than in the air or upper sand layers, and they affect the formation of the local Mediterranean coast microclimate [67]. During the period of January to March, when the seawater temperatures were ≤9.9 °C, *Aspergillus* species prevailed in sand. However, *Aspergillus* and *Penicillium* numbers were the lowest in months when seawater is warmer, the air pressure is higher, the wind is stronger, and the precipitation is ≤105.7 mm^3^. Air pressure is the variable that has a broad effect on weather and corresponds to the changes in wind speed and direction, as well as the rainy and sunny periods. Higher air pressure indicates sunny weather and lower air pressure is associated with precipitation. On the contrary, the above-described factors favor the growth of *Aphanoascus*, *Fusarium*, *Bisifusarium*, and *Rhizopus*, which is in accordance with previous studies, reporting elevated presence of these genera in warm and/or humid climate conditions [66,68,69,70]. Besides weather, also, the antagonism between *Aspergillus* and *Penicillium* versus *Fusarium* and *Bisifusarium* could affect the ratio of these genera in the sand [71]. 

### 4.4. Fungi in Sand as Possible Indicators of Anthropogenic Pollution during the Bathing Season

Another focus was the recognition of possible fungal indicators of human pollution, which appeared mainly during the bathing season. Indicator microorganisms appear after a certain cascade of events, e.g., *Escherichia coli* and enterococci indicate faecal pollution in the environment [15], while the data on fungal indicators are scarce. With the culturable approach we identified four fungal genera, namely *Actinomucor*, *Condenascus*, *Meyerozyma*, and *Stachybotrys*, that may serve also as potential human pollution indicators during the bathing season. Their numbers at the beach sand started to increase from May to October and reached the peak from June to September. Machine learning analysis yielded the air temperature >28 °C during the past 7 days before sampling as the most decisive factor for the presence of all four genera in the sand. The presence of the species *Condenascus tortuosus* is predicted to be the most numerous among all indicator fungi. It was isolated from May to September when the air temperatures are either higher than 28 °C or when T ≤ 28 °C and the air pressure is ≤1012.0 mbar (indicating warm and moist weather). This genus was described recently and the available data currently links it only to plants and soil, but not to human health [72]. Although isolated under the same cascade of environmental factors as *Condenascus*, the allergenic and mycotoxigenic fungus *Stachybotrys* was predicted in lowest numbers among indicator fungi. Its presence in beach sand was reported in many other studies, showing beach sand as a potential reservoir for this genus [13]. *Actinomucor elegans* is a ubiquitous fungus, related mainly to plants, soil, and insects. Machine learning predicts its presence in the beach sand mainly during the spring and summer months with higher air pressure (indicating sunny and warm weather), which can be related to plant growth and insects’ activity [62]. This species was recently added to the list of emerging clinically relevant fungi [73]; thus, its elevated presence in sand during the bathing season should not be neglected. Although all indicator genera were present during the bathing season, only the yeast *Meyerozyma* can be linked to the presence of humans, since it was isolated during the summer and autumn months with sunny, warm or hot weather, associated with the highest numbers of beach visits [13]. This result confirms previous observations on sand and consolidates the *Meyerozyma* species as a possible fungal indicator of human faecal pollution [13].

Besides possible indicators of anthropogenic pollution, the sporadic presence of thermotolerant species such as *Arachniotus flavoluteus*, *Geotrichum* spp., *Mucor circinelloides*, *Phialophora americana*, *Trichoderma citrinoviride*, and *Trichosporon asahii* was detected during the bathing season. These are known to cause mild to severe fungal infections, and should be taken into consideration if their numbers in beach sand elevate [34].

### 4.5. Beach Sand Harbours Amphotericin B-Resistant Strains of Aspergillus Section Nigri

Although beach sand harboured highly diverse mycobiota, the most numerous fungi belong to the genus *Aspergillus*. Nine species were isolated from the beach sand during the monitoring, while only four were present during the official bathing season. Interestingly, *Aspergillus fumigatus* was not isolated, although its presence is well documented in sand [12,13,51]. Only one strain of its sibling species, *A. lentulus*, was isolated in October after the flood event. Representatives of the genus *Aspergillus* can cause allergies, sinusitis, otitis, keratitis, but also life-threatening infections, such as invasive aspergillosis [34]. In addition, CDC takes it under scrutiny due to the fast development of resistance to antifungals, not only in hospitals, but also in the natural environment [74,75]. The isolates obtained during the bathing season were tested against nine antifungals, and the obtained results were compared with the recent EUCAST data [37,76] and the data from de Hoog et al. [34]. Echinocandines are not recommended clinical treatment against *Aspergillus*, and, as previously reported, *Aspergillus* strains showed aberrant growth endpoint after 48 h (in our case >8 µg/mL) [36,37,77]. In our study, MIC could not be determined for fluconazole, in accordance with EUCAST data [37]. EUCAST often reports intrinsic resistance for *A. flavus*, *A. nidulans*, and *A. niger* wild types against posaconazole, voriconazole, and itraconazol, but all strains from beach sand were susceptible to these antimycotics [37]. *Aspergillus flavus* and *A. nidulans* showed similar result against amphotericin B as previously reported for wild types (ECOFF = 4 µg/mL), but *Aspergillus niger* and its close relative *A. welwitschiae* (section Nigri) were resistant to this antimycotic with MIC = 2 µg/mL. ECOFF for *A. niger* is currently set between 0.5 and 1 µg/mL, while MIC values greater than 1 µg/mL are considered as strain’s resistance. Although the tests were carried out on a small number of isolates, the observed resistance for wild types of *Aspergillus niger* and possible resistance of *A. welwitschiae* against amphotericin B should be taken into consideration in further studies of beach sand.

## 5. Conclusions

To contribute to the knowledge on the mycobiota of sand and water, the present study investigated not only the abundance and diversity of the culturable fraction of the fungal community, but also the fluctuations of the most abundant genera through the year, and the possible fungal indicators for the bathing season. The data were analyzed with machine learning (ML) methods that can combine all factors screened, identifying combinations of the most important factors favoring certain fungal groups. The obtained ML models identify the weather conditions, i.e., air and water temperature, sunshine hours, precipitation, humidity, air pressure, and wind speed, as the key factors influencing sand and sea mycobiota. In addition, and for the first time, culturable methods yielded four genera that could be followed as potential human pollution indicators during the bathing season. Amongst them, and based on findings from other worldwide studies, we suggest the genus *Meyerozyma* as a candidate to be monitored as a human pollution indicator. Attention should be paid also to the whole-year present *Aspergillus* spp. strains. Resistance against amphotericin B was found in *A. niger* and possibly also in *A. welwitschiae*. Due to their role in fungal otitis, especially in children, this should encourage future studies on resistance in *Aspergillus* strains, isolated from beach sand and sea. Altogether, our findings lay the foundation for further research on sand and seawater mycobiota and suggest the potential effect of weather changes on the presence of opportunistic and resistant fungi in sand and sea. Diversity analyses based on sequencing methods could be used in the future to provide further insights into environmental fungal diversity and comparisons between beach sand and seawater.

## Figures and Tables

**Figure 1 jof-08-00860-f001:**
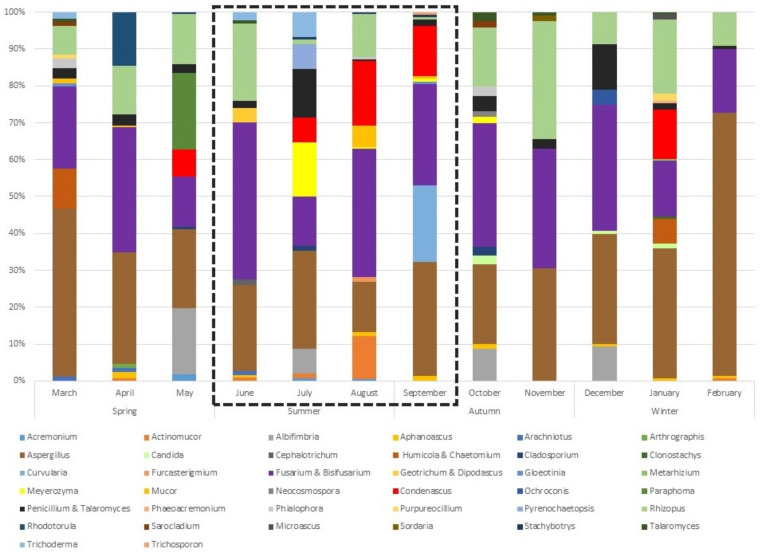
Relative abundance of cultured fungal genera (in %) in the samples taken each month during the period of 12 months. The official bathing season is indicated with black dashed square.

**Figure 2 jof-08-00860-f002:**
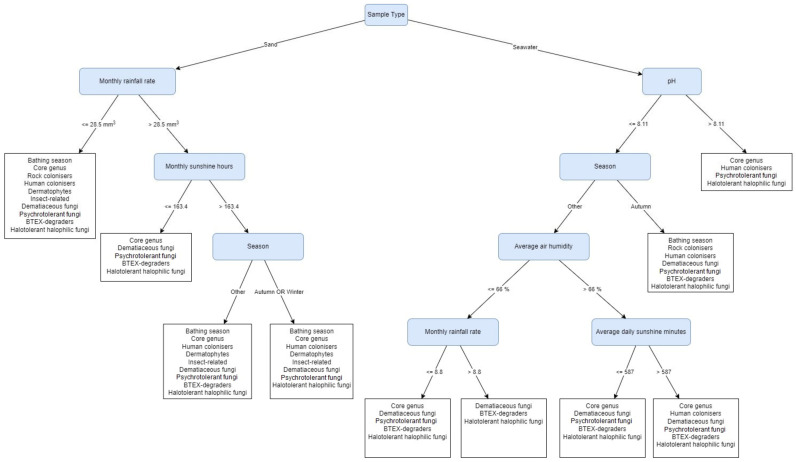
The Predictive Clustering Tree (PCT) for multi-label classification (MLC), predicting the presence of fungal functional groups, isolated from beach sand and seawater. The decision process starts by separating samples based on their type (sand or seawater). From there on, Monthly rainfall rate, Monthly sunshine hours, Season, pH, Average air humidity and Average daily sunshine minutes are used to determine the prediction of the tree. For example, if samples come from seawater and their pH is higher than 8.11, the predicted labels (functional groups) are: Core genus, Human colonisers, Psychrotolerant fungi, and Halotolerant/halophilic fungi.

**Figure 3 jof-08-00860-f003:**
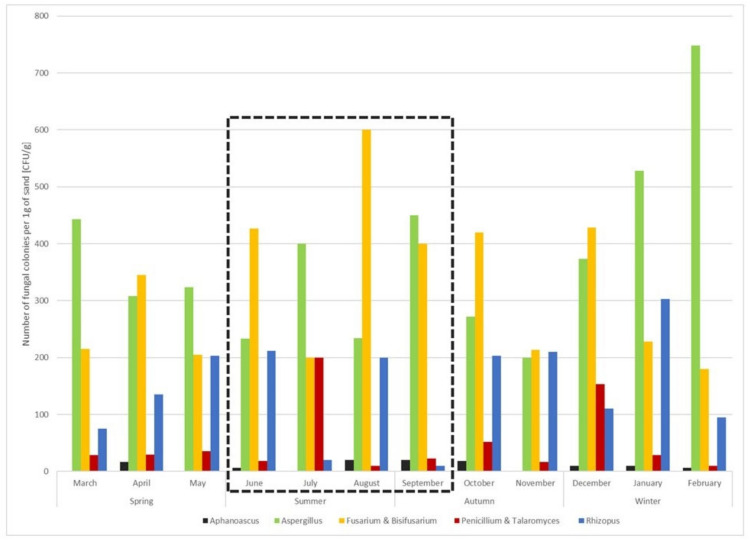
Presence and numbers of “core-genera” in sand samples during the period of 12 months. The official bathing season is indicated with a black, dashed square.

**Figure 4 jof-08-00860-f004:**
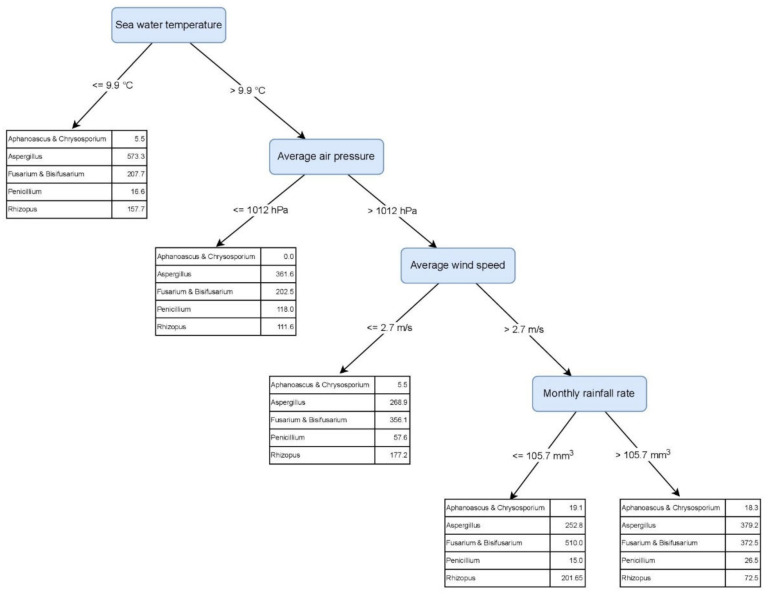
Predictive Clustering Tree (PCT) for multi-target regression (MTR), predicting the abundance of the core-genera, isolated from beach sand and seawater. The model predicts the highest abundances for *Aspergillus* and *Fusarium & Bisifusarium*. Particularly high abundance of *Aspergillus* is predicted when sea water temperature is below or equal to 9.9 °C (leftmost leaf), while high abundance of *Fusarium & Bisifusarium* is predicted when sea water is warmer (>9.9 °C), air pressure is high (>1012 hPa), average wind speeds are higher than 2.7 m/s and monthly rainfall rate is below or equal to 105.7 mm^3^ (rightmost leaf). The genus *Aphanoascus* is always predicted in lowest quantities.

**Figure 5 jof-08-00860-f005:**
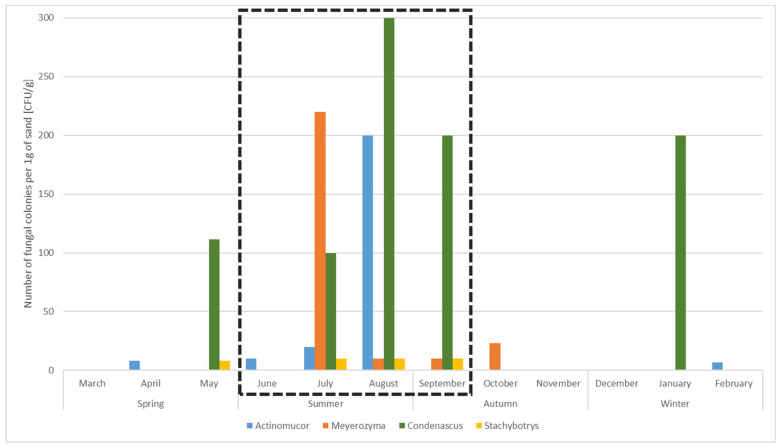
Presence and numbers of “potential human pollution indicator-genera” in sand samples during the period of 12 months. The official bathing season is indicated with a black, dashed square.

**Figure 6 jof-08-00860-f006:**
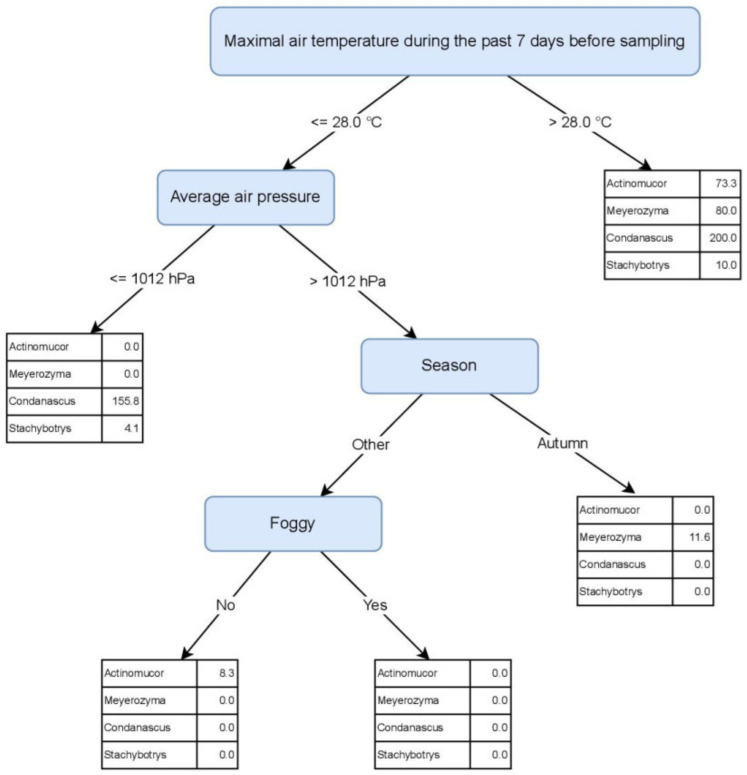
Predictive Clustering Tree (PCT) for multi-target regression (MTR), predicting the abundances of potential human pollution indicators, isolated from beach sand and seawater. For the case when the Maximal air temperature during the past 7 days before sampling is higher than 28 °C, the predicted abundances (CFU/g) are given in the top-most leaf node of the tree (*Actinomucor* = 73.3 CFU/g, *Meyerozyma* = 80.0 CFU/g, *Condenascus* = 200.0 CFU/g, *Stachybotrys* = 10.0 CFU/g). If temperature is less than or equal to 28 °C, another test is applied on the Average air pressure attribute. This way, the decision tree is traversed and every sample eventually ends up in one of the leaf nodes of the tree, where predictions for fungal abundances are provided.

**Table 1 jof-08-00860-t001:** Fungal species isolated from beach sand and seawater in Portorož, Slovenia during the monitoring.

Identification	Genetic Barcode	EXF-No. ^1^	GenBank No.	Month	Habitat	Biosafety Level(BSL) ^5^	Other Characteristics or Role in Habitats ^6^
*Acremonium masseei*	ITS	EXF-14878	MT280604	August ^2,3^	BS	No data	colonising plants, soil, freshwater
*Acremonium* sp.	ITS	EXF-14657	MT280605	July ^2,3^	BS	BSL-1	colonising plants, soil, freshwater
*Actinomucor elegans*	ITS	EXF-14384 EXF-14449 EXF-14621 EXF-14845 EXF-14849 EXF-14639	MT280606 MT280607 MT280608 MT280609 MT280610 MT280611	February, April, June ^2,3^, July ^2,3^, August ^2,3^	BS	BSL-2	colonising plants, soil, freshwater, insect-related, potential human pollution indicator
*Albifimbria verrucaria*	ITS	EXF-14006 EXF-14159 EXF-14548 EXF-14648	MT280612 MT280613 MT280614 MT280615	May ^3^, July ^2,3^, October ^4^, December	BS	BSL-1	colonising plants, soil, freshwater
*Alternaria* sp.	ITS	EXF-13999	MT280685	October^4^	SW	BSL-1	colonising plants, soil, freshwater
*Aphanoascus fulvescens*	ITS	EXF-14459 EXF-14870 EXF-14871	MT280616 MT280617 MT280618	April, September ^2,3^	SW	BSL-2	dermatophyte, “core-genus” in the sand
*Aphanoascus reticulisporus*	ITS	EXF-14632	MT280619	June ^2,3^	BS	BSL-1	dermatophyte,“core-genus” in the sand
*Aphanoascus terreus*	ITS	EXF-13894 EXF-14308 EXF-14461	MT280627 MT280628MT280629	January, April, October ^4^	BS	BSL-1	dermatophyte,“core-genus” in the sand
*Arachniotus flavoluteus*	ITS	EXF-14420 EXF-14458 EXF-14628	MT280620 MT280621 MT280622	March, April, June ^2,3^	BS	BSL-1	dermatophyte
*Arthrographis curvata*	ITS	EXF-14460	MT280623	April	BS	BSL-1	dermatophyte
*Aspergillus calidoustus*	*benA*	EXF-14640	MT328464	July ^2,3^	BS	BSL-1	dematiaceous, halotolerant, “core-genus” in the sand, colonising plants, soil, freshwater
*Aspergillus destruens*	*benA*	EXF-14429	MT328468	February	SW	BSL-1	dematiaceous, halotolerant, “core-genus” in the sand, colonising plants, soil, freshwater
*Aspergillus flavus*	*benA*	EXF-13896 EXF-14002 EXF-13898 EXF-13905 EXF-13907 EXF-14163 EXF-14011 EXF-14285 EXF-14296 EXF-14383 EXF-14389 EXF-14404 EXF-14862	MT328469 MT328470 MT328471 MT328472 MT328473 MT328474 MT328475 MT328476 MT328477 MT328478 MT328479 MT328480 MT328481	January, February, March, May ^3^, June ^2,3^, July ^2,3^, August ^2,3^, September ^2,3^, October ^4^, November, December	BS	BSL-2	dematiaceous, halotolerant, “core-genus” in the sand, colonising plants, soil, freshwater
*Aspergillus floccosus*	*benA*	EXF-14455 EXF-14615	MT328455 MT328456	April, June ^2,3^	BS	BSL-1	dematiaceous, halotolerant, “core-genus” in the sand, colonising plants, soil, freshwater
*Aspergillus jensenii*	*benA*	EXF-14541 EXF-14542	MT328465 MT328466	May^3^	SW	BSL-1	dematiaceous, halotolerant, “core-genus” in the sand, colonising plants, soil, freshwater
*Aspergillus lentulus*	ITS	EXF-13910	MT280624	October^4^	BS	BSL-2	dematiaceous, halotolerant, “core-genus” in the sand, colonising plants, soil, freshwater
*Aspergillus nidulans*	*benA*	EXF-14003 EXF-13911 EXF-14014 EXF-14167 EXF-14298 EXF-14388 EXF-14392 EXF-14393 EXF-14405 EXF-14410 EXF-14452 EXF-14643 EXF-14850 EXF-14854 EXF-14465	MT328430 MT328431 MT328432 MT328433 MT328434 MT328435 MT328436 MT328437 MT328438 MT328439 MT328440 MT328441 MT328442 MT328443 MT328444	January, February, March, July ^2,3^, August ^2,3^, September ^2,3^, October ^4^, November, December	BS	BSL-1	dematiaceous, halotolerant, “core-genus” in the sand, colonising plants, soil, freshwater
*Aspergillus niger*	*benA*	EXF-14015 EXF-14297 EXF-14558 EXF-14618 EXF-14873	MT328457 MT328458 MT328459 MT328460 MT328461	January, May ^3^, June ^2,3^, August ^2,3^, November	BS	BSL-1	dematiaceous, halotolerant, “core-genus” in the sand, colonising plants, soil, freshwater
*Aspergillus pseudoglaucus*	*benA*	EXF-14397	MT328467	March	SW	BSL-1	dematiaceous, halotolerant, “core-genus” in the sand, colonising plants, soil, freshwater
*Aspergillus terreus*	*benA*	EXF-14294 EXF-14387 EXF-14863	MT328445 MT328446 MT328447	January, February, September^2,3^	BS	BSL-2	dematiaceous, halotolerant, “core-genus” in the sand, colonising plants, soil, freshwater
*Aspergillus tubingensis*	*benA*	EXF-14165 EXF-14289 EXF-14381 EXF-14453 EXF-14464 EXF-14555 EXF-14543	MT328448 MT328449 MT328450 MT328451MT328452 MT328453 MT328454	January, February, April, May^3^, December	BS, SW	BSL-1	dematiaceous, halotolerant, “core-genus” in the sand, colonising plants, soil, freshwater
*Aspergillus welwitchiae*	*benA*	EXF-14400 EXF-14864	MT328462 MT328463	March, September ^2,3^	BS	No data	dematiaceous, halotolerant, “core-genus” in the sand, colonising plants, soil, freshwater
*Aureobasidium melanogenum*	ITS	EXF-13886	MT280686	October^4^	SW	BSL-1	dematiaceous, halotolerant, BTEX degrader, colonising plants, soil, freshwater
*Aureobasidium pullulans*	ITS	EXF-14611 EXF-14638	MT280687 MT280688	June ^2,3^, July ^2,3^	SW	BSL-1	dematiaceous, halotolerant, BTEX degrader, colonising plants, soil, freshwater
*Bisifusarium delphinoides*	*tef1*	EXF-14645 EXF-14166	MT292641 MT292642	July ^2,3^, December	BS	BSL-1	“core-genus” in the sand, colonising plants, soil, freshwater
*Candida parapsilosis*	LSU	EXF-13884 EXF-13882 EXF-13881 EXF-14282 EXF-14300	MT273264 MT273265 MT273266 MT273267 MT273268	October ^4^, December, January	BS	BSL-1	colonising humans, colonising plants, soil, freshwater
*Candida tropicalis*	LSU	EXF-14299	MT273269	January	BS	BSL-2	colonising humans, colonising plants, soil, freshwater
*Cephalotrichum gorgonifer*	ITS	EXF-14630	MT280625	June ^2,3^	BS	BSL-1	rock-inhabiting
*Chaetomidium fimeti*	ITS	EXF-14301	MT280626	January	BS	BSL-1	rock-inhabiting, dematiaceous, colonising plants, soil, freshwater
*Cladosporium cladosporioides*	*act*	EXF-13913 EXF-13912 EXF-14378	MT292643 MT292644 MT292645	February, October ^4^	SW	BSL-1	dematiaceous, psychrotolerant, halotolerant, BTEX degrader, colonising plants, soil, freshwater
*Cladosporium halotolerans*	*act*	EXF-14442 EXF-14443 EXF-14004 EXF-14562	MT292646 MT292647 MT292648 MT292649	April, May ^3^	BS, SW	BSL-1	dematiaceous, psychrotolerant, halotolerant, BTEX degrader, colonising plants, soil, freshwater
*Cladosporium pseudocladosporioides*	*act*	EXF-13897 EXF-14379	MT292650 MT292651	February, October^4^	SW	BSL-1	dematiaceous, psychrotolerant, halotolerant, BTEX degrader, colonising plants, soil, freshwater
*Cladosporium ramotenellum*	*act*	EXF-13916 EXF-14380	MT292652 MT292653	February, October^4^	SW	BSL-1	dematiaceous, psychrotolerant, halotolerant, BTEX degrader, colonising plants, soil, freshwater
*Cladosporium sphaerospermum*	*act*	EXF-14540	MT292656	May^3^	SW	BSL-1	dematiaceous, psychrotolerant, halotolerant, BTEX degrader, colonising plants, soil, freshwater
*Cladosporium tenellum*	*act*	EXF-14001	MT292654	October^4^	SW	BSL-1	dematiaceous, psychrotolerant, halotolerant, BTEX degrader, colonising plants, soil, freshwater
*Cladosporium velox*	*act*	EXF-13914	MT292655	October ^4^	SW	BSL-1	dematiaceous, psychrotolerant, halotolerant, BTEX degrader, colonising plants, soil, freshwater
*Clonostachys rosea*	ITS	EXF-14306	MT280630	January	BS	BSL-1	insect-related, colonising plants, soil, freshwater
*Condenascus tortuosus*	ITS	EXF-14287 EXF-14553 EXF-14554 EXF-14653 EXF-14844 EXF-14858	MT280631 MT280632 MT280633 MT280634 MT280635 MT280636	January, May ^3^, July ^2,3^, August ^2,3^, September ^2,3^	BS	No data	indicator in bathing season, colonising plants, soil, freshwater
*Curvularia* sp.	ITS	EXF-14855 EXF-14856	MT280637 MT280638	September ^2,3^	BS	BSL-1	dematiaceous, colonising plants, soil, freshwater
*Cystofilobasidium macerans*	LSU	EXF-14427	MT273278	February	SW	BSL-1	psychrotolerant, halotolerant
*Didymella glomerata*	ITS	EXF-14613	MT280689	June ^2,3^	SW	BSL-1	dematiaceous, colonising plants, soil, freshwater
*Dipodascus geotrichum*	ITS	EXF-14631 EXF-14624	MT280640 MT280641	June ^2,3^	BS	BSL-1	colonising humans, plants
*Exophiala xenobiotica*	ITS	EXF-13891	MT280690	October ^4^	SW	BSL-1	rock-inhabiting, dematiaceous, BTEX degrader, colonising humans, freshwater
*Furcasterigmium furcatum*	ITS	EXF-14876	MT280639	August ^2,3^	BS	BSL-1	rock-inhabiting, colonising plants, soil, freshwater
*Fusarium brachygibbosum*	*tef1*	EXF-14451	MT292640	April	BS	BSL-1	“core-genus” in the sand, colonising plants, soil, freshwater
*Fusarium ipomoeae*	*tef1*	EXF-14157	MT292639	December	BS	BSL-1	“core-genus” in the sand, colonising plants, soil, freshwater
*Fusarium nirenbergiae*	*tef1*	EXF-13888 EXF-14164 EXF-14385 EXF-14399 EXF-14411 EXF-14447 EXF-14448 EXF-14547 EXF-14549 EXF-14626 EXF-14641 EXF-14860 EXF-14843	MT292623 MT292624 MT292625 MT292626 MT292627 MT292628 MT292629 MT292630 MT292631 MT292632 MT292633 MT292634 MT292635	February, March, April, May ^3^, June ^2,3^, July ^2,3^, August ^2,3^, September ^2,3^, October ^4^, December	BS	BSL-1	“core-genus” in the sand, colonising plants, soil, freshwater
*Fusarium solani*	*tef1*	EXF-14848 EXF-14005 EXF-14288	MT292636 MT292637 MT292638	January, August ^2,3^, October ^4^	BS	BSL-2	“core-genus” in the sand, colonising plants, soil, freshwater
*Geotrichum silvicola*	ITS	EXF-14629	MT280642	June ^2,3^	BS	BSL-1	colonising humans, colonising plants, soil, freshwater
*Gliomastix roseogrisea*	ITS	EXF-14564	MT280643	May ^3^	BS	BSL-1	BTEX degrader, dematiaceous, insect-related, colonising plants, soil, freshwater
*Gloeotinia* sp.	ITS	EXF-14407 EXF-14868	MT280644 MT280645	March, September ^2,3^	BS	No data	psychrotolerant, colonising plants, soil, freshwater
*Hortaea werneckii*	ITS	EXF-14000	MT280691	October ^4^	SW	BSL-1	dematiaceous, halophilic
*Humicola homopilata*	ITS	EXF-14398	MT280646	March	BS	No data	rock-inhabiting, dematiaceous, colonising plants, soil, freshwater
*Lenzites betulinus*	ITS	EXF-14377	MT280692	February	SW	No data	colonising plants, soil, freshwater
*Metarhizium marquandii*	ITS	EXF-14305	MT280647	January	BS	BSL-1	insect-related, colonising plants, soil, freshwater
*Meyerozyma caribbica*	LSU	EXF-14647 EXF-14650 EXF-14655 EXF-14881 EXF-14867 EXF-13879	MT273270 MT273271 MT273272 MT273273 MT273274 MT273275	July ^2,3^, August ^2,3^, September ^2,3^, October ^4^	BS	BSL-1	colonising humans, colonising plants, soil, freshwater, potential human pollution indicator
*Meyerozyma guilliermondii*	LSU	EXF-13885	MT273279	October ^4^	SW	BSL-1	colonising humans, colonising plants, soil, freshwater, potential human pollution indicator
*Microascus brevicaulis*	ITS	EXF-14307	MT280648	January	BS	BSL-2	colonising plants, soil, freshwater
*Mucor circinelloides*	ITS	EXF-14403 EXF-14853 EXF-14450 EXF-14847	MT280649 MT280650 MT280651 MT280652	March, April, August ^2,3^, September ^2,3^	BS	BSL-1	colonising plants, soil, freshwater
*Neocosmospora rubicola*	ITS	EXF-13893	MT280653	October ^4^	BS	No data	colonising plants, soil, freshwater
*Neomicrosphaeropsis italica*	ITS	EXF-14074	MT280693	November	SW	No data	colonising plants, soil, freshwater
*Ochroconis* sp.	ITS	EXF-14160	MT280654	December	BS	BSL-1	dematiaceous, BTEX degrader, colonising plants, soil, freshwater
*Papiliotrema fonsecae*	LSU	EXF-14426	MT273280	February	SW	BSL-1	halotolerant
*Paraconiothyrium cyclothyrioides*	ITS	EXF-13917 EXF-14007 EXF-14614	MT280694 MT280695 MT280696	June ^2,3^, October ^4^, November	SW	BSL-1	halotolerant
*Paraphoma radicina*	ITS	EXF-14550	MT280655	May ^3^	BS	BSL-1	dematiceaous, colonising plants, soil, freshwater
*Penicillium aurantiogriseum*	*benA*	EXF-13906 EXF-14386 EXF-14402 EXF-14644	MT328501 MT328502 MT328503 MT328504	February, March, July ^2,3^, October ^4^	BS	BSL-1	psychrotolerant, halotolerant, “core-genus” in the sand, colonising plants, soil, freshwater
*Penicillium crustosum*	*benA*	EXF-14646 EXF-14444	MT328495 MT328496	April, July ^2,3^	BS, SW	BSL-1	psychrotolerant, halotolerant, “core-genus” in the sand, colonising plants, soil, freshwater
*Penicillium glabrum*	*benA*	EXF-14169	MT328513	December	BS	BSL-1	psychrotolerant, halotolerant, “core-genus” in the sand, colonising plants, soil, freshwater
*Penicillium griseofulvum*	*benA*	EXF-13899 EXF-14170 EXF-14172 EXF-14156 EXF-14158 EXF-14423 EXF-14552 EXF-14563	MT328505 MT328506 MT328507 MT328508 MT328509 MT328510 MT328511 MT328512	March, May ^3^, October ^4^, December	BS	BSL-1	psychrotolerant, halotolerant, “core-genus” in the sand, colonising plants, soil, freshwater
*Penicillium oxalicum*	*benA*	EXF-14292 EXF-14445	MT328493 MT328494	January, April	BS	BSL-1	psychrotolerant, halotolerant, “core-genus” in the sand, colonising plants, soil, freshwater
*Penicillium sizovae*	*benA*	EXF-14880	MT328498	August ^2,3^	BS	BSL-1	psychrotolerant, halotolerant, “core-genus” in the sand, colonising plants, soil, freshwater
*Penicillium* sp.	*benA*	EXF-14610	MT328497	June ^2,3^	SW	BSL-1	psychrotolerant, halotolerant, “core-genus” in the sand, colonising plants, soil, freshwater
*Phaeoacremonium iranianum*	ITS	EXF-14302	MT280656	January	BS	No data	colonising plants, soil, freshwater
*Phialophora americana*	ITS	EXF-14409 EXF-14879	MT280659 MT280660	March, August ^2,3^	BS	BSL-2	dematiaceous, BTEX degrader, colonising plants, soil, freshwater
*Phialophora* sp.	ITS	EXF-13890 EXF-14073	MT280657 MT280658	October ^4^	BS	BSL-1	dematiaceous, colonising plants, soil, freshwater
*Purpureocillium* sp.	*benA*	EXF-14303 EXF-14421	MT328499 MT328500	January, March	BS	BSL-1/-2	rock-inhabiting, halotolerant, colonising plants, soil, freshwater
*Pyrenochaetopsis tabarestanensis*	ITS	EXF-14642	MT280661	July ^2,3^	BS	No data	colonising plants, soil, freshwater
*Rhizopus arrhizus*	ITS	EXF-13903 EXF-13909 EXF-14009 EXF-14012 EXF-14168 EXF-14290 EXF-14291 EXF-14293 EXF-14390 EXF-14454 EXF-14556 EXF-14627 EXF-14649 EXF-14846 EXF-14852	MT280662 MT280663 MT280664 MT280665 MT280666 MT280667 MT280668 MT280669 MT280670 MT280671 MT280672 MT280673 MT280674 MT280675 MT280676	January, February, April, May ^3^, June ^2,3^, July ^2,3^, August ^2,3^, September ^2,3^, October ^4^, November, December	BS	BSL-2	“core-genus” in the sand, colonising plants, soil, freshwater
*Rhodotorula graminis*	LSU	EXF-14539	MT273282	January	SW	BSL-1	psychrotolerant, halotolerant, BTEX-degrader, colonising plants, soil, freshwater
*Rhodotorula mucilaginosa*	LSU	EXF-14446	MT273276	April	BS	BSL-1	halotolerant, BTEX-degrader, colonising human, plants, soil, freshwater
*Sarocladium kiliense*	ITS	EXF-13880 EXF-14408	MT280677 MT280678	March, October^4^	BS	BSL-2	insect-related, colonising plants, soil, freshwater
*Sordaria nodulifera*	ITS	EXF-14010	MT280679	November	BS	No data	colonising plants, soil, freshwater
*Stachybotrys* sp.	ITS	EXF-14565 EXF-14654 EXF-14875	MT280680 MT280681 MT280682	May ^3^, July ^2,3^, August ^2,3^, September ^2,3^	BS	BSL-1	colonising plants, soil, freshwater, potential human pollution indicator
*Stereum hirsutum*	ITS	EXF-13915	MT280697	October^4^	SW	BSL-1	colonising plants, soil, freshwater
*Talaromyces atroroseus*	*benA*	EXF-14286	MT328482	January	BS	BSL-1	insect-related, colonising plants, soil, freshwater, “core-genus” in the sand
*Talaromyces pinophilus*	*benA*	EXF-14406 EXF-13900 EXF-14017 EXF-13904 EXF-13902 EXF-14415 EXF-13901 EXF-14620 EXF-14633	MT328483 MT328484 MT328485 MT328486 MT328487 MT328488 MT328489 MT328490 MT328491	March, May ^3^, October ^4^, November	BS	BSL-1	insect-related, colonising plants, soil, freshwater, “core-genus” in the sand
*Talaromyces purpureogenus*	*benA*	EXF-13895	MT328492	October ^4^	BS	BSL-1	insect-related, colonising plants, soil, freshwater, “core-genus” in the sand
*Trichoderma citrinoviride*	ITS	EXF-14412 EXF-14619	MT280683 MT280684	March, June ^2,3^	BS	BSL-1	colonising plants, soil, freshwater
*Trichosporon asahii*	LSU	EXF-14865	MT273277	September ^2,3^	BS	BSL-2	dermatophyte
*Wickerhamomyces anomalus*	LSU	EXF-14842	MT273283	August ^2,3^	SW	BSL-2	colonising humans, plants, soil, freshwater

Legend: ^1^ EXF No., number designated to fungi in the EX Culture Collection of the Infrastructural Centre Mycosmo, Biotechnical Faculty, University of Ljubljana, Slovenia. ^2^ Official bathing season. ^3^ Beach cleaning, mechanical aeration. ^4^ A one-time event, flood. ^5^ Biosafety Level (BSL) data cited from de Hoog et al. [34] and ATCC [35]. ^6^ Fungal characteristics/role in habitats cited from Brandão et al. [13], de Hoog et al. [34] and Mycoses Study Group Education and Research Consortium, 2022 (https://drfungus.org/ (accessed on 16 August 2021)). BS—beach sand. SW—seawater.

**Table 2 jof-08-00860-t002:** The results of susceptibility testing for Aspergillus species isolated during bathing season against nine antifungals from Sensititre Yeast One Kit.

Identification	EXF-No.	Range of Minimal Inhibitory Concentration (MIC) [µg/mL]								
		Anidulafungin (AND), 24 h	Micafungin (MF), 24 h	Caspofungin (CAS), 24 h	Amphotericin B (AB), 48 h	5-Flucytosine (FC), 48 h	Posaconazole (PZ), 48 h	Voriconazole (VOR), 48 h	Itraconazole (IZ), 48 h	Fluconazole (FZ), 48 h
*Aspergillus flavus*	EXF-14617	0.03 ^4^	0.06 ^4^	1 ^4^	2–4	>64 ^4^	0.06 ^4^	0.5–1	0.06 ^4^	≥256 ^4^
	EXF-14651									
	EXF-14874									
	EXF-14862									
	ECOFF ^1^	IE ^2^	IE ^2^	IE ^2^	4	no data	0.5	2	1	/ ^3^
*Aspergillus nidulans*	EXF-14643	0.03 ^4^	0.12 ^4^	0.25 ^4^	2–4	>64 ^4^	0.06–0.12	0.12–0.25	0.03–0.06	≥256 ^4^
	EXF-14850									
	EXF-14854									
	ECOFF ^1^	IE ^2^	IE ^2^	IE ^2^	4	no data	0.5	1	1	/ ^3^
*Aspergillus niger*	EXF-14618	0.03 ^4^	0.12 ^4^	1 ^4^	2 ^4^	32 ^4^	0.12 ^4^	1 ^4^	0.25 ^4^	≥256 ^4^
	EXF-14873									
	ECOFF ^1^	IE ^2^	IE ^2^	IE ^2^	0.5–1	no data	0.25–0.5	2	4	/ ^3^
*Aspergillus welwitschiae*	EXF-14864	0.015	0.06	0.5	2	16	0.12	0.5	0.12	≥256
	ECOFF ^1^	no data	no data	no data	no data	no data	no data	no data	no data	no data

Legend: ^1^ ECOFF—Epidemiologic cut-off values for wild-type strains, as abbreviated by EUCAST [37]. ^2^ Insufficient evidence that the organism or group is a good target for therapy with the agent [37]. ^3^ No breakpoints. Susceptibility testing is not recommended [37]. ^4^ All tested strains had the same result; the numbers represent the average value for all strains of the same species. Red colour—resistant; growth of the strain was above ECOFF reported for the species [37]. Pink colour—possibly resistant; growth of the strain was above ECOFF reported for the taxonomically closest relative *Aspergillus niger* [37]. Green colour—susceptible; growth of the strain was below ECOFF reported for the species [37].

## Data Availability

Not applicable.

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
