# Peer review of "Occurrence, Diversity and Anti-Fungal Resistance of Fungi in Sand of an Urban Beach in Slovenia—Environmental Monitoring with Possible Health Risk Implications"

_jof, 2022, doi:10.3390/jof8080860_

Round 1
Reviewer 1 Report
The manuscript: “Occurrence, diversity and anti-fungal resistance of fungi in sand of an urban beach in Slovenia – environmental monitoring with possible health risk implications” by Monika Novak Babič, Nina Gunde-Cimerman, Martin Breskvar, Sašo Džeroski, and João Brandã, is well written in general and provide insights about a culturable fraction of fungal diversity in seawater and beach sand from a popular beach in Slovenia. The research is original and interesting for the public, and may be considered as a first reference for future studies. Importantly, the research was able to detect candidate fungal species as indicators of human and environmental pollution.
Major comments
- Write scientific names properly, in italics for genera and species. Please correct in your manuscript if that's the case. I am not sure if this could be a manner of the journal system during submission.
- Please indicate, in the final part of the introduction, that the study is limited to culture-specific approaches for the isolation and characterization of fungi. There are many fungal species in the beach sand that we are not aware about how to culture them yet, or many culture media and techniques may be required to truly recover most of the fungal diversity. To date, only diversity approaches considering amplicon sequencing, metagenomics and other omics, and imaging techniques could provide further insights about the extent of environmental diversity of fungi.
- I strongly suggest reviewing the materials and methods sentences and paragraphs that correspond mostly to results and information for the discussion. I recommend following the methods style for the writing of this section. Also, there are repeated phrases including information in methods and discussion: for example the blue flag sentence. I include some example paragraphs in minor comments.
- In the results: “A one year-long survey with monthly sampling of beach sand yielded (delete: a great) great insights into the diversity of (include: culturable) culturable fungal species (suggestion: integrating instead of forming) forming the fungal community/ that may be candidates for fungal monitoring of human pollution, for example...
- For section 3.2: (please include Culturable) fungal (delete community, include diversity) community (so: culturable fungal diversity) in seawater has little in common with the community in beach sand. I suggest: Differences in culturable fungal diversity in seawater and beach sand. Remember you are only analyzing a fraction of the fungal diversity, limited to the culture techniques and media.
- For the conclusions section, my suggestion is to indicate that you are only analyzing a fraction of the fungal diversity, the culturable fraction limited to your media and techniques, but it was sufficient to detect potential species for contamination monitoring and with inexpensive reagents (compared to sequencing techniques for example) and classic microbiology techniques. Also, I suggest to include that diversity analyses based on sequencing methods could provide further insights into environmental fungal diversity and comparisons between beach sand and sea water in the future. Sequencing approaches are increasingly getting cheaper. I strongly suggest continuing your research considering amplicon sequencing of fungal marker genes in the future.
Minor comments
- I suggest including also microorganisms in the sentence: “beach sand contains organic matter, originating from -include microorganisms- plants, animals, human activities, and the sea”.
- In materials and methods, can you provide more information about the cleaning methods of the beach? It is about pick up trash by machines or people, or other more chemical or detergent methods?.
- You mention that the beach under study was created artificially, do you know the origin of the sand used for the task or any other information that may be relevant to better understand the nature of the sand?.
- This paragraph should be included in the results, not in the methods section: “Fungi of the genus Aspergillus were in general the most commonly isolated species from the beach sand samples […] in order to evaluate possible risk for humans during their leisure activities on the beach”.
- Adjust the paragraph for section 2.1 for methods style. Some sentences can also be moved to results.
- Section 2.5 also contains sentences that better correspond to results and discussion.
- Please capitalize and write as: World Health Organization.
- Suggestion: “Amongst them, and based on findings from other worldwide studies, we suggest the genus Meyerozyma to be monitored as a –potential or candidate- human pollution indicator”. More studies may be needed to characterize the isolates, there is a rich diversity of environmental and host-associated microbes that require detailed characterization of their gene repertoirs.
Author Response
Please see the attachment with point by point response.

Reviewer 2 Report
The topic of the paper is ecologically interesting and important. However, the methodology used and the results' interpretation raise some questions and doubts, which need to be clarified.
Abstract. Hereafter throughout the text – all genus and species names must be written in italic.
Materials and Methods.
Lines 88-89: This sentence belongs to the end of Introduction, but not to M&M.
Line 99: why namely from this depth and not from the surface?
Subsection 2.5. It is an important detail – why have namely these parameters been chosen? Why are they such extremely (on my mind – unnecessarily) numerous? Why do they include both numerical characteristics (temperature, humidity, rainfall amount, etc.) and categorical ones (season, type of weather, events, etc.)? Why do they include many covariate characteristics (for example, type of weather is determined by temperature, the amount of sunshine hours, etc.)? How was the sand differentiated on "wet, humid, and dry" and why was its moisture content or water holding capacity not just measured in the lab? Howbeit, all these parameters should be presented in a table per month of sampling (possibly, in supplementary materials) to give the reader a comprehensive picture of the environmental conditions.
Subsections 2.6 and 2.7. On my mind, such extremely detailed description of the modeling approaches using the highly specific terminology hardly understandable for the readers without specific statistical background is unnecessary for a mycological paper.
Results.
Subsection 3.1. Why did the authors concentrate on the variations in genera and not – in species, which, on my mind, is more important?
Subsection 3.3.
For me, the separation of studied sand fungal community into 11(?!) functional/non-taxonomic groups looks rather problematic. Almost all isolated fungi can be associated with much more than one group, some of fungal functional activities are known to be strain- but not species-specific. Such definition as "fungi related to plants (parasites? endophytes? epiphytes?), soil, and freshwater" is extremely broad and can be applied to almost all species. The presence of "psychrotolerant and psychrophiles" group looks strange taking into account that the species were isolated at 25, 30 and 37°C, while maximal growth temperature for psychrophilic fungi is known to be 20°C. Much more logical would be to separate the group of thermotolerant fungi, which are able to grow at 37°C; being able to grow at the temperature of human body, these fungi may create a potential risk for human health, and it is especially relevant to aspergilli. Additionally, it is not clear what was the source based on which the authors related the isolated fungi to a particular group. Also because of all these concerns, the results presenting in Figure 2 are not convincing for me.
Table 1 is extremely long - 17 pages(!!). I am sure that the strain information (EXF No and GenBank No.) can be easily placed in a separate table in supplementary materials. The authors should really check the species names with the Index Fungorum database in order to avoid using the inappropriate species names (see the PDF file of the manuscript).
Subsection 3.4.
Line 360: Penicillium and Talaromyces are two different genera in the modern fungal taxonomy.
Subsection 3.6. There is no need to repeat in the text the information contained in Table 2.
In the whole, the use of machine learning approach as a tool for relating fungal composition to environmental conditions does not seem fully convincing for me. Maybe the major problem is associated with the extremely numerous and largely covariate environmental conditions applied, and because of that the information presenting in the corresponding figures rather blurs the real relationship between the fungal community and its environment instead of shedding light on it. At least, the authors should explain why such important and direct factors as temperature and moisture of the sand do not influence the studied characteristics of fungal communities, while such very indirect factor as air pressure – does. Additionally, it is not clear for me why all these trees are predictive (what do they predict?) and not just descriptive.
Discussion is hardly readable mainly because it unnecessarily repeats (the first paragraph – almost fully) what has been already written in Introduction, M&M, and Results.
Some explanations for the variations in fungal composition look rather weak. I can hardly believe in sunshine hours, being "one of the most influencing for microbial communities in sand" (lines 569-571) – only as a fact that the sunshine hours influence air temperature and it in turn – the temperature of substrate (sand). Besides that, why was the lowest diversity registered in November and not in December, with really lowest sunshine hours?
Conclusions should contain only the conclusive remarks on own research but not the general statements as on lines 658-663. It is rather strange that the first and only mention of global warming and extreme whether events (what does it mean?) appears only at the end of Conclusions.
All other corrections and suggestions are inserted into the attached PDF file of the manuscript.

Author Response
Reviewer 2
Comments and Suggestions for Authors
The topic of the paper is ecologically interesting and important. However, the methodology used and the results' interpretation raise some questions and doubts, which need to be clarified.
We thank Reviewer 2 for the thorough evaluation of the manuscript including a careful check of the text and the taxonomical data. It has helped us to considerably improved our work.
Abstract
Hereafter throughout the text – all genus and species names must be written in italic.
The errors in the fonts used occurred during the merging of formats in the JoF template. We have now italicised the names of all fungal genera and species.
Materials and Methods.
Lines 88-89: This sentence belongs to the end of Introduction, but not to M&M.
The sentence in question, i.e., “The goal was to broadly investigate fungi in beach sand and recreational waters [13].” has now been removed from the text. Instead, the following sentence has been added: “Sampling was conducted at the Central beach in Portorož, Slovenia, as a part of the wider European initiative “Mycosands” [13].” The remainder of the paragraph was moved to the end of the Introduction, as suggested by the reviewer.
Line 99: why namely from this depth and not from the surface?
All participating countries of the Mycosands initiative followed a pre-arranged Standard Operational Procedure (SOP). The SOP was adopted from a previous study conducted by Sabino et al [19] and agreed among the team leaders of the Mycosands project (Brandao et al., 2021 – [13]).
Subsection 2.5.
It is an important detail – why have namely these parameters been chosen? Why are they such extremely (on my mind – unnecessarily) numerous?
While the presence of fungi in sand is well known, very little is known about the environmental factors that influence the mycobiota of sand. We wanted to identify the most important (single or combined) factors influencing the presence and abundance of fungi in beach sand and seawater, thus the idea of collecting as much data (parameters) as possible and analysing it with an approach that enables such broad cross-comparison (machine learning).
Why do they include both numerical characteristics (temperature, humidity, rainfall amount, etc.) and categorical ones (season, type of weather, events, etc.)?
The parameters include both types, numerical and categorical data, because any of them might influence the mycobiota of sand and seawater. In fact, as described in the manuscript, also single events like a flood or the New Year celebration significantly changed the diversity and abundance of fungi. The chosen machine learning method (Predictive Clustering Trees - PCTs) is capable of exploiting numerical, categorical or both kinds of attributes simultaneously when learning the model.
Why do they include many covariate characteristics (for example, type of weather is determined by temperature, the amount of sunshine hours, etc.)?
Covariate characteristics contribute also data on complementary, indirectly influencing factors. For example, the type of weather includes temperature and sunshine hours. While temperature is a single parameter, sunshine hours also give indirect data about the impact of UV-radiation. It is not the same if the month is warm (high temperatures), but cloudy (less sunshine hours, and consequently less UV reaches the ground) or the month is both warm (high temperatures) and sunny (more sunshine hours, and consequently more UV reaches the ground).
Typically, a handful of attributes are determined / selected in advance, i.e., before the data collection. Then, traditional statistical approaches are used for hypothesis testing. When the number of attributes grows, so does the complexity of analysis, especially when using traditional statistical approaches. However, with machine learning approaches, the number of data attributes can be large. The relations between the attributes can be direct or indirect, weak or strong and the attributes need not be independent of each other. We rely on the machine learning (ML) algorithm to find the best predictive model. Since the goal of predictive modeling is to build a model that is able to accurately predict a value, the models themselves can expose interesting relations between attributes that are not necessarily always in line with currently known relations. This does not mean that the current / existing knowledge is not valid in our study, it just means that our data supports an alternate view (exposed via ML).
How was the sand differentiated on "wet, humid, and dry" and why was its moisture content or water holding capacity not just measured in the lab?
We agree with Reviewer 2, but as for sampling depth, all participating countries of the Mycosands initiative followed a pre-arranged Standard Operational Procedure (SOP) adopted from a previous study conducted by Sabino et al [19], where sand was differentiated descriptively.
The sand was described as wet (coastal bottom sediment), when it was completely soaked (e.g. during or closely after rain periods), humid (vadose zone sand) when it held the structure during the sampling, and dry (supra-tidal sand) when its structure was completely loose.
Howbeit, all these parameters should be presented in a table per month of sampling (possibly, in supplementary materials) to give the reader a comprehensive picture of the environmental conditions.
We agree with Reviewer 2 and have now provided a table with all collected parameters during the monitoring in a Supplement.
Subsections 2.6 and 2.7. On my mind, such extremely detailed description of the modeling approaches using the highly specific terminology hardly understandable for the readers without specific statistical background is unnecessary for a mycological paper.
We have completely removed the part of the text in subsection 2.6 Predictive modelling shown below:
“Predictive models can be seen as mathematical functions that take inputs and produce outputs. There are many predictive model types and most of them are black-box models, i.e., models that make predictions that are (very) difficult to explain. Such models often exhibit competitive predictive performance, but are not particularly useful when explanations are required. In those cases, so-called white-box model types can be used. One such model type are decision trees.”
While we understand the concern of Reviewer 2, we would like to keep the remaining parts of Machine Learning method descriptions. We believe they might help interested readers to understand the background of Machine Learning, since this kind of methods are to date not widely used in mycological studies, yet can give a fresh view on data interpretation.
Results.
Subsection 3.1. Why did the authors concentrate on the variations in genera and not – in species, which, on my mind, is more important?
We decided to focus on the genera rather than species in order to have better statistical robustness. With the number of species we isolated during the study, the transparency in Figure 1 would be completely lost. However, Subsection 3.1. includes also Table 1 with all data on species level, habitat and the month of isolation.
Subsection 3.3.
For me, the separation of studied sand fungal community into 11(?!) functional/non-taxonomic groups looks rather problematic. Almost all isolated fungi can be associated with much more than one group, some of fungal functional activities are known to be strain- but not species-specific. Such definition as "fungi related to plants (parasites? endophytes? epiphytes?), soil, and freshwater" is extremely broad and can be applied to almost all species. The presence of "psychrotolerant and psychrophiles" group looks strange taking into account that the species were isolated at 25, 30 and 37°C, while maximal growth temperature for psychrophilic fungi is known to be 20°C. Much more logical would be to separate the group of thermotolerant fungi, which are able to grow at 37°C; being able to grow at the temperature of human body, these fungi may create a potential risk for human health, and it is especially relevant to aspergilli. Additionally, it is not clear what was the source based on which the authors related the isolated fungi to a particular group. Also because of all these concerns, the results presenting in Figure 2 are not convincing for me.
We understand the Reviewer 2’s concerns regarding the data presentation in Figure 2. Figure 2 was primarily designed to show the connection between isolated mycobiota and environmental factors. Machine learning was used to construct two models. One by using the the MTR approach (predicting abundances) and the other by the MLC approach (predicting presence/ absence). In both cases, the model type used were Predictive clustering trees (PCTs). We tested both approaches before we finally decided to use functional/ecological groups instead of taxonomical hierarchy.
To explain:
- At first, we used the taxonomical hierarchy/ groups, where we assigned each species to the genus, order and phylum. When we used such data in the machine learning analysis, the results were inconclusive and the model prediction for the whole community was too weak.
- When we used functional/ecological groups as already proposed and published by Brandao et al. (2021), we usually assigned more than one role to each species. Such "classification" allows putting together different, genetically non-related taxa, which in nature coexist under certain cascades of events. Such grouping gave much better model predictions for the isolated fungal community.
The data processing:
As Reviewer 2 mentioned, also the Machine Learning (ML) approach recognised the group "fungi related to plants, soil, and freshwater" as assigned to almost every genus in the analysis. The result of the ML analysis was: Fungi, classified under the label "fungi related to plants, soil, and freshwater" were isolated from sand samples regardless of the environmental factors.
Unlike taxonomical classification, the labelling into ecological groups allowed us to assign more than one label to each genus. That enabled us to analyse fungal genera with multiple labels. While the label "fungi related to plants, soil, and freshwater" was excluded from the analyses, genera with more than just this label were analysed under the remaining labels.
As Reviewer 2 suggests, we agree on removing the word “psychrophilic fungi” and leave only “psychrotolerant fungi”, which have a growth spectrum from 0 – 30°C.
During the analysis we indeed made also a group of fungi labelled as “Thermotolerant fungi”, but the data pool was too small and too scattered to have an impact on the generated model. As for the fungi, classified under the label "fungi related to plants, soil, and freshwater", the result of such analysis was: “Thermotolerant fungi” were isolated from sand samples regardless of the environmental factors.
The fungal characteristics / roles in habitats were extracted from Brandão et al. [13], de Hoog et al. [34] and Mycoses Study Group Education and Research Consortium, 2022 (https://drfungus.org/).
The following item was thus added to the Legend:
6 Fungal characteristics / role in habitats as adopted from Brandão et al. [13], de Hoog et al. [34] and Mycoses Study Group Education and Research Consortium, 2022 (https://drfungus.org/)
Table 1 is extremely long - 17 pages(!!). I am sure that the strain information (EXF No and GenBank No.) can be easily placed in a separate table in supplementary materials.
Although we agree with Reviewer 2 on the length of Table 1, we would like to maintain it in the current form. The table is large due to the number of isolated and identified species, which was the aim of the study. We have generated the paper under the support of the Infrastructural Center Mycosmo with the largest global collection of extremophiles (EX), thus removing EXF-Numbers and assigned GenBank Numbers could have a negative impact on the distribution and of this collection and the publicity/ attention it receives.
The authors should really check the species names with the Index Fungorum database in order to avoid using the inappropriate species names (see the PDF file of the manuscript).
We thank Reviewer 2 for his/her help on taxonomical data and classification. All suggestions were followed and the names were corrected accordingly.
Subsection 3.4.
Line 360: Penicillium and Talaromyces are two different genera in the modern fungal taxonomy.
We thank Reviewer 2 for his/her help on taxonomical data and classification. All suggestions were followed and the names were corrected accordingly.
Subsection 3.6. There is no need to repeat in the text the information contained in Table 2.
Corrected according to Reviewer 2’s suggestions.
In the whole, the use of machine learning approach as a tool for relating fungal composition to environmental conditions does not seem fully convincing for me. Maybe the major problem is associated with the extremely numerous and largely covariate environmental conditions applied, and because of that the information presenting in the corresponding figures rather blurs the real relationship between the fungal community and its environment instead of shedding light on it.
Please, see also the answers discussing the machine learning methods above. In general, ML methods are designed to be used on data of different sizes, both in terms of the number of samples and the number of attributes in order to model the relation of the independent variables (descriptive and/or numerical) and the dependent ones and highlight the most important combinations of factors that lead to the occurrence of certain fungal groups.
It is the first time that this kind of method has been used on beach sand samples, which gives us the very first insight in the combinations of factors influencing fungal occurrence/abundance. Currently, the Mycosands Initiative has started the second European sampling and we hope we will be able to compare and confirm the generated models in the future in order to find stronger connections between fungal biota and weather events.
At least, the authors should explain why such important and direct factors as temperature and moisture of the sand do not influence the studied characteristics of fungal communities, while such very indirect factor as air pressure – does.
Reviewers 2’s statement cannot be applied for all analyses, because the results depend on the selected output data. For instance, Figure 2 highlights the monthly rainfall rate, sunshine hours and average air humidity as the most important variables influencing the presence of fungal groups (output data) in the sand and sea.
When studying only the core-genera (output data) (Figure 4), seawater temperature was recognised as the most decisive factor, followed by the air pressure. Air pressure is the variable which includes also the changes in wind speed and direction, and is directly connected to the rainy and sunny periods (higher air pressure – more likely sunny weather, lower air pressure - changes in wind and precipitation). We included this explanation in the discussion as: “However, Aspergillus and Penicillium numbers were the lowest in months when seawater gets warmer, the air pressure is higher, the wind is stronger, and the precipitation is ≤105.7 mm3. Air pressure is the variable which has a broad effect on weather and corresponds to the changes in wind speed and direction, as well as the rainy and sunny periods. Higher air pressure indicates sunny weather and lower air pressure is associated with precipitation. On the contrary, the above-described factors favor the growth of Aphanoascus, Fusarium, Bisifusarium, and Rhizopus, which is in accordance with previous studies, reporting elevated presence of these genera in warm and / or humid climate conditions [66,68-70].”
In Figure 6, predicting the abundances of potential human pollution indicators (output data), the first decisive factor is the maximal air temperature during the last 7 days before sampling, followed by the air pressure. Focusing on Meyerozyma, that would mean the highest chance (and abundance) to isolate it from the sand when:
- the maximal air temperature during the last 7 days before sampling is higher than 28°C,
- but it still can be isolated also during the autumn months, when T<28°C and the average air pressure is >1012 hPa (indicating sunny (not necessarily hot) weather and consequently more people on the beach).
Additionally, it is not clear for me why all these trees are predictive (what do they predict?) and not just descriptive.
Besides being descriptive, the models/tree are also predictive, because they predict the presence and/or abundance of selected fungal groups from combinations of environmental factors.
- Figure 2. The Predictive Clustering Tree (PCT) for multi-label classification (MLC), predicting the presence of fungal functional groups, isolated from beach sand and seawater.
- Figure 4. Predictive Clustering Tree (PCT) for multi-target regression (MTR), predicting the abundance of the core-genera, isolated from beach sand and seawater.
- Figure 6. Predictive Clustering Tree (PCT) for multi-target regression (MTR), predicting the abundances of potential human pollution indicators, isolated from beach sand and seawater.
Discussion is hardly readable mainly because it unnecessarily repeats (the first paragraph – almost fully) what has been already written in Introduction, M&M, and Results.
The Discussion and the other sections were corrected as suggested by Reviewer 2. Please, see the attached manuscript with track changes.
Some explanations for the variations in fungal composition look rather weak. I can hardly believe in sunshine hours, being "one of the most influencing for microbial communities in sand" (lines 569-571) – only as a fact that the sunshine hours influence air temperature and it in turn – the temperature of substrate (sand).
As discussed above, sunshine hours do not influence only the temperature of the air and ground, but they provide also indirect data about UV-exposure. Sunny days are strongly associated with the higher UV radiation (A, B and C), which is known to have an impact on the microbial communities in the water and sand.
We have added the following text in this part of the discussion to clarify the variation: “The likely reason for lower diversity in November is the drop of monthly sunshine hours. This factor was previously reported as one of the most influencing for microbial communities in sand due to its double effect on daily temperatures and UV-irradiation [12,13].”
Besides that, why was the lowest diversity registered in November and not in December, with really lowest sunshine hours?
We observed that anomaly and we connected the elevated presence of fungi in sand during December and January to the open celebrations that took place on the urban beach. When sampling during these months, we namely observed a lot of paper and plastic waste that likely contributed to the isolation of human-related fungal genera.
In the paper, we recommend recording any unexpected / unusual events (flood, heat wave, exceptionally strong wind, construction work, open celebration, etc.), since they can have an impact on fungal diversity / abundance.
Conclusions should contain only the conclusive remarks on own research but not the general statements as on lines 658-663. It is rather strange that the first and only mention of global warming and extreme whether events (what does it mean?) appears only at the end of Conclusions.
The following part of the conclusion has been completely deleted, as suggested by Reviewer 2: ”Fungal presence in the beach sand of urban and natural beaches has been increasingly investigated during the last few years, and is frequently connected to plants, plant debris, wild animals, and the composition of the sand itself. Yet, possible connections between the presence of certain fungi in sand and seawater and human health is severely understudied. Microbial safety regulation of beaches is still based on traditional faecal indicators, leaving out fungal opportunists causing ailments in susceptible beach users.”
In addition, the last sentence has been corrected and now reads: “Altogether, our findings lay the foundation for further research on sand and seawater mycobiota and suggest the potential effect of weather changes on the presence of opportunistic and resistant fungi in sand and sea.”
All other corrections and suggestions are inserted into the attached PDF file of the manuscript.
We thank Reviewer 2 again for his/her extensive work in providing valuable comments and suggestions. All comments in the PDF file were checked and addressed. All corrections can be found in the MS Word file of the revised manuscript, where they are clearly indicated via the track changes option.

Round 2
Reviewer 2 Report
The authors responded to all my questions and concerns, nevertheless, some of the explanations have not convinced me.
Concerning the set of environmental parameters involving into the analysis – I still think that they are excessively numerous. I can understand the desire of authors to relate the composition of sand and seawater fungal communities to maximal number of environmental parameters via the ML approach. However, this approach as well as other statistical methods is just a computer program, and it is the responsibility of researcher to input into the program the relevant parameters in order to obtain the logically explainable output. The logic in this particular case is that the studied fungal communities inhabit the precise environment – sand, and namely its characteristics – moisture, temperature, salinity (probably changing during the year due to the changing moisture), aeration, pH, organic matter (probably increasing during the bathing season), etc. , more or less directly influence the communities' formation, while the involving climatic parameters are very indirect influencers.
The supplementary table for the environmental parameters looks strange with all these data repetitions. The first column "sample" with "sand and sea water" should be removed, and only the columns containing different data for sand and sea water like "pH" should be divided into two sub-columns. And what does the column "sea water" mean?
On my mind, the description of ML methodology (subsections 2.6 - 2.9) is still too detailed. I understand that this methodology is rather exotic unlike the conventional statistical approaches like multiple regression, RDA, CCA, etc. for numerical data and a several-way ANOVA with interactions for categorical data, but even such, this description can be shorten taking into account that the figures presenting the ML results also contain the description of methodology. The interested readers will still have to refer to the original description of the approach. They will be convinced to apply namely this methodology in their research by the strength and relevance of the results it gives rather than by its detailed description.
I still strongly believe, based on my more than 40-yeared experience in studying soil fungi that the species-specific analysis of fungal communities whenever it is possible is more informative than the genus-specific (lines 342-347) irrespective of statistical robustness (the fact that the ML figures are based on the cumulative abundance of genera is another story). It is especially relevant for the analysis of core sand community (subsection 3.4) and human pollution indicators (subsection 3.5). Noting that the genera Penicillium and Aspergillus belong to the so called "core genera" gives little information because these genera are the richest among soil fungi and are consistently present in almost all (if not in all) types of soil. It is much more interesting and important to display (not only in the table but in the text as well) and analyze the abundance of concrete species from these genera as well as from others.
I fully agree with the authors that the analysis of functional groups containing the species with different ecological-functional preferences is more relevant for the study purposes than the analysis of taxonomic groups, once again irrespective of statistical robustness. However, the fact that the content of functional groups in this particular case is overlapping to great extent, diminishes the results of such analysis. Moreover, the range of isolation temperatures does not allow the authors to be sure that they isolated namely psychrotolerant but not just mesophilic species. And once again irrespective of statistical robustness, the authors should discuss the thermotolerance of isolated species because being able to grow at the temperature of human body, these fungi may create a potential risk for human health, and it is especially relevant for aspergilli, which were constituted to be dominant in the studied sand communities.
Concerning Table 1 – I suggested not to remove the strain information (EXF No and GenBank No.) at all, but just to place it in a separate table in the Supplementary Materials, which would make Table 1 readable.
Concerning the predictable character of the ML trees – maybe the models do predict the presence/abundance of groups/genera, but Figure 4, for example, shows not the predictive but the obtained abundance of the isolated genera.
Other comments:
Line 177: the authors should explain the choice namely of this depth for the isolation of fungi as they explained it in the response to reviewer comments.
Lines 224-225: the authors should explain how the sand was differentiated on "wet, humid, and dry", as they explained it in the response to reviewer comments - at least, in the footnote to the supplementary table.
Subsection 3.5. Figure 3 shows the increase in abundance of Fusarium&Bisifusarium on July – so why are these paired genera not considered "potential human pollution indicators"?
Table 2. Once again, - whether the values are averages – it should be specified.
Line 807: but good aeration is beneficial for fungi - so why "even"?
Line 881: rather weak explanation taking into account that pH of the substrate was alkaline during the whole period of study.
Line 939: concerning UV, it is also a weak explanation, because UV radiation is known to penetrate only up to about 100 microns into soil depth (Johnson, 2003).
Lines 1033-1039: this part is not a discussion, but just the result repetition.
Nevertheless, I admit that some of my concerns may be rather subjective. Overall, the paper contains some interesting findings and after minor corrections, it can be considered for publication.
All other corrections and suggestions are inserted into the attached PDF file of the manuscript.
Author Response
Authors sincerely thank to Reviewer 2 for his/her time and detailed revision of the manuscript. His/her long years of working experience in a similar field have helped us to improve our manuscript.
Please see the detailed response in the attachment.
